# Metabolic but not transcriptional regulation by PKM2 is important for natural killer cell responses

Jessica F Walls[1,2], Jeff J Subleski[2], Erika M Palmieri[2], Marieli Gonzalez-Cotto[2], Clair M Gardiner[1†], Daniel W McVicar[2†], David K Finlay[1,3†]*

[1]School of Biochemistry and Immunology, Trinity Biomedical Sciences Institute, Trinity College Dublin, Dublin, Ireland; [2]Laboratory of Cancer Immunometabolism, National Cancer Institute, Frederick, United States; [3]School of Pharmacy and Pharmaceutical Sciences, Trinity Biomedical Sciences Institute, Trinity College Dublin, Dublin, Ireland

**Abstract** Natural Killer (NK) cells have an important role in immune responses to viruses and tumours. Integrating changes in signal transduction pathways and cellular metabolism is essential for effective NK cells responses. The glycolytic enzyme Pyruvate Kinase Muscle 2 (PKM2) has described roles in regulating glycolytic flux and signal transduction, particularly gene transcription. While PKM2 expression is robustly induced in activated NK cells, mice lacking PKM2 in NK cells showed no defect in NK cell metabolism, transcription or antiviral responses to MCMV infection. NK cell metabolism was maintained due to compensatory PKM1 expression in PKM2-null NK cells. To further investigate the role of PKM2, we used TEPP-46, which increases PKM2 catalytic activity while inhibiting any PKM2 signalling functions. NK cells activated with TEPP-46 had reduced effector function due to TEPP-46-induced increases in oxidative stress. Overall, PKM2-regulated glycolytic metabolism and redox status, not transcriptional control, facilitate optimal NK cells responses.

*For correspondence: finlayd@tcd.ie

†These authors contributed equally to this work

Competing interests: The authors declare that no competing interests exist.

## Introduction

Natural Killer (NK) cells are key lymphocytes for the control of viral infection and cancer immunosurveillance. Glycolysis is important for the function of NK cells as inhibition of glycolysis disrupts normal NK cell effector functions including the production of IFNγ and the lysis of target cells (*Assmann et al., 2017*; *Donnelly et al., 2014*; *Keating et al., 2016*; *Loftus et al., 2018*; *Kedia-Mehta et al., 2019*). Glucose breakdown through glycolysis provides energy, in the form of ATP, both directly and via generation of pyruvate to fuel mitochondrial oxidative phosphorylation (OXPHOS). Oxidation of glucose also produces glycolytic intermediates that feed into ancillary metabolic pathways such as the pentose phosphate pathway (PPP) to support cellular processes including biosynthesis and antioxidant activities. For instance, the glycolytic intermediate glucose-6-phosphate can feed into the PPP for the production of nucleotides and to generate NADPH, an important cofactor for cellular biosynthesis and for maintaining cellular redox control. Glycolytic intermediates and enzymes can also play a role in directly regulating immune functions. For example, in T lymphocytes, the metabolite phosphoenolpyruvate can control $Ca^{2+}$/NFAT signalling and the glycolytic enzyme glyceraldehyde-3 phosphate dehydrogenase (GAPDH) has a role outside of glycolysis where it can control the translation of IFNγ and IL-2 mRNAs (*Ho et al., 2015*; *Chang et al., 2013*). Another glycolytic enzyme linked to the control of immune functions is an isoform of the final enzyme in glycolysis, pyruvate kinase muscle (PKM).

In most tissues, alternative splicing of the PKM gene yields two isoforms, PKM1 and PKM2. PKM1 forms a homo tetramer that efficiently converts phosphoenolpyruvate and ADP into pyruvate and ATP. While PKM2 can similarly form a catalytically efficient tetramer, it is also found as a monomer/dimer (called monomeric PKM2 hereafter) that has substantially less catalytic activity (*Christofk et al., 2008*). Therefore, in situations where PKM2 is predominantly mono/dimeric, the rate of glycolytic flux may be slowed leading to an accumulation of upstream glycolytic intermediates that can be diverted into other pathways such as the PPP to support cellular biosynthesis. Accordingly, PKM2 expression has been found to be elevated in many cells with high biosynthetic burdens including tumour cells (*Li et al., 2018*; *Hitosugi et al., 2009*). Monomeric PKM2 has also been shown to have functions independent of its metabolic role in glycolysis, primarily in the nucleus where it contributes to transcriptional control through interactions with the hypoxia inducible factor-1α (HIF-1α) and signal transduction and activator of transcription (STAT) transcription factors (*Hitosugi et al., 2009*; *Demaria and Poli, 2012*; *Dong et al., 2016*). The balance between catalytically less active mono/dimeric PKM2, with potential signalling roles, and catalytically more active tetrameric PKM2, without signalling roles, is controlled by a range of cellular factors. For instance, mono/dimeric PKM2 is converted to the catalytically efficient tetramer when levels of upstream metabolites such as fructose-1,6-phosphate or serine are high (*Keller et al., 2012*; *Morgan et al., 2013*; *Anastasiou et al., 2012*). In this way, PKM2 operates as a rheostat by sensing upstream pools of glycolytic and biosynthetic intermediates and tuning glycolytic rates accordingly to carefully balance metabolic needs for cellular growth and function.

While activated NK cells are known to engage elevated levels of glycolysis to support rapid growth, proliferation and function, how glycolytic flux is controlled to promote these processes in NK cells is not understood. In particular, what role PKM2 plays in supporting NK cell metabolic and functional responses is unknown. Here, we use genetic and pharmacological approaches in vitro to show that PKM2 does not have a significant role in regulating the transcriptional landscape of NK cells. Additionally, we find that although PKM2 is highly expressed in activated NK cells, in the absence of PKM2 these cells can precisely adjust the expression of PKM1 to control overall PKM activity demonstrating a remarkable capability of NK cells for precise metabolic plasticity. In contrast, acute pharmacological activation of PKM2 catalytic activity blunted NK cell growth and effector functions. These defects were associated with increased levels of ROS and a transcriptional signature indicating oxidative stress that we linked to reduced PPP metabolites and decreased NADPH levels. Overall, this study reveals an important metabolic role for PKM2 in supporting the PPP and redox balance in activated NK cells and highlights that PKM2 does not have a profound transcriptional role in all immune cells.

## Results

### PKM2 expression is induced in activated NK cells

To investigate whether NK cell activation in vivo is associated with changes in PKM2 expression mice were injected with poly(I:C) and splenocytes were isolated for analysis. There was a significant increase in the expression of PKM2 in NK cells from mice that received poly(I:C) as measured by flow cytometry (*Figure 1a*). To further investigate PKM2 expression splenic NK cells were cultured for 6 days in low-dose IL-15, a cytokine required for DC-mediated NK cell priming in vivo (*Lucas et al., 2007*; *Dubois et al., 2002*; *Koka et al., 2004*) (called 'cultured NK cells' hereafter). These cultured NK cells were purified and stimulated with IL-2 plus IL-12 and PKM2 protein and mRNA expression measured over the course of 24 hr by western blot analysis and rtPCR, respectively. The expression of PKM2 increased over time following activation peaking at 24 hr post cytokine stimulation (*Figure 1b*). The mammalian Target of Rapamycin complex 1 (mTORC1) has been shown to be active both in NK cells stimulated in vivo following poly(I:C) injection, and in IL-2/IL-12 stimulated NK cells, and is an important regulator of NK cell metabolism (*Donnelly et al., 2014*). Inhibition of mTORC1 activity using the inhibitor rapamycin significantly reduced the abundance of PKM2 mRNA and protein in IL-2/12 stimulated cultured NK cells (*Figure 1c*). Rapamycin efficacy was confirmed by immunoblot for phosphorylation of the mTORC1 target, ribosomal s6 protein (*Figure 1c*). Similarly, we confirmed that *Pkm1* mRNA is expressed and regulated by mTORC1 in IL-2/12 stimulated NK cells along with PKM1 protein (*Figure 1—figure supplement 1a–b*). Although we were primarily

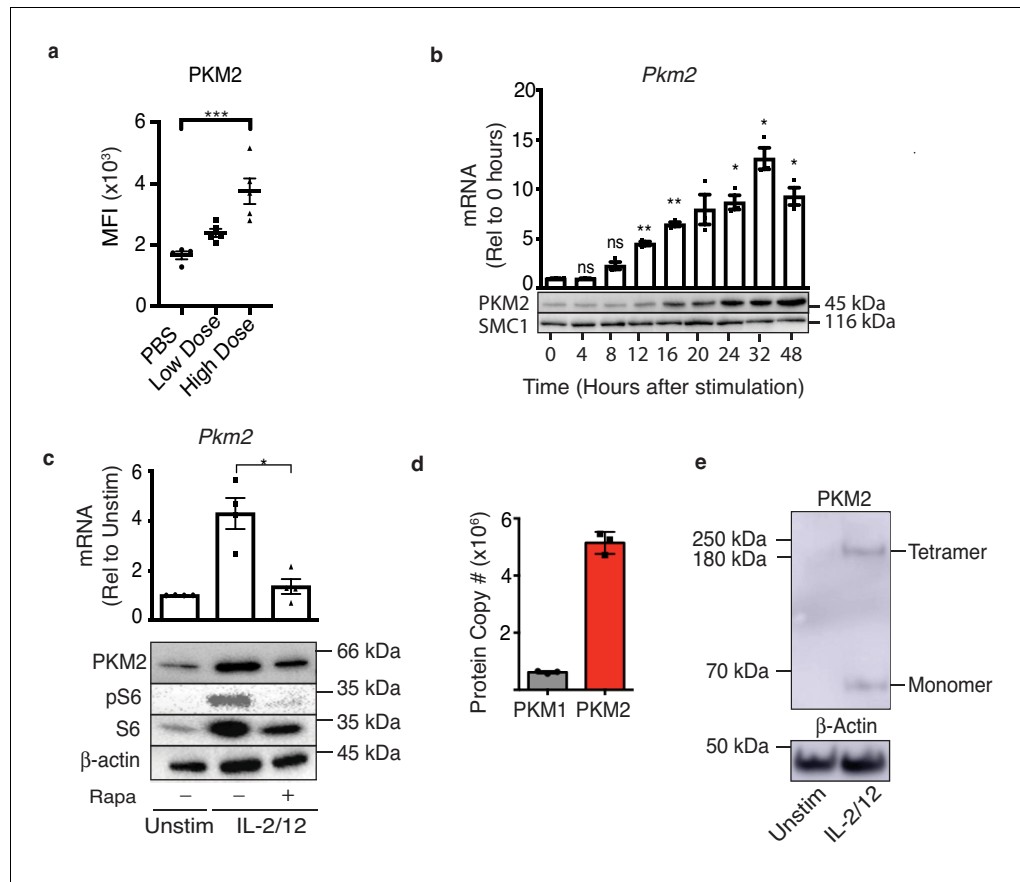

**Figure 1.** PKM2 is expressed and is the predominant PKM isoform in activated murine NK cells. (a) Wildtype C57Bl/6 mice were injected with saline (100 µL), low-dose poly(I:C) (100 µg/100 µL) or high-dose poly(I:C) (200 µg/ 100 µL) I. P. Spleens were harvested 24 hr post-injection and PKM2 expression was analysed by intracellular flow cytometry in NK1.1[+] NKp46[+] cells (b) NK cell cultures were activated with IL-2/12 for 48 hr and cells were lysed for protein and mRNA. Samples were analysed by immunoblot for PKM2 and SMC1 protein expression. mRNA samples were subjected to qPCR analysis and *Pkm2* expression over time was determined relative to time zero. Data was normalised to housekeeping gene *Rplp0*. (c) Cultured NK cells were stimulated for 18 hr in IL-2/12 +/- rapamycin. After 18 hr cells were harvested for protein and mRNA. Samples were analysed by immunoblot for PKM2, β-Actin, total S6 and pS6. mRNA samples were subjected to qPCR analysis for *Pkm2* expression. Data was normalised to housekeeping gene *Rplp0*. (d) Levels of individual peptides for PKM1 and PKM2 were compared using quantitative proteomics. **a** Data are mean +/- S.E.M for 4–5 mice per group in two individual experiments. (**b–e**) Data were analysed using one-way ANOVA with Tukey post-test and are pooled or representative of three individual experiments. [*]p>0.05, [**]p>0.01, [***]p>0.001.

The online version of this article includes the following figure supplement(s) for figure 1:

**Figure supplement 1.** PKM1 expression is increased with IL-2/12 stimulation.

concerned with the pyruvate kinase muscle (PKM) family of enzymes, we also confirmed that NK cells do not express the liver- and erythrocyte-specific isoforms of pyruvate kinase, PK-L and PK-R respectively, using quantitative proteomics. We observed upwards of $6 \times 10^6$ copies of PKM peptide in IL-2/12 stimulated NK cells, with no detectable peptides from the *Pklr* gene (*Figure 1—figure supplement 1c*). As both PKM1 and PKM2 are induced with IL-2/12 stimulation, we next asked if they are present in comparable quantities. Interestingly, quantitative proteomics analysis revealed that PKM2 is ninefold more abundant than PKM1 in cytokine activated NK cells (*Figure 1d*). PKM2 is known to exist in three different oligomeric conformations, monomers, dimers and tetramers (*Angiari et al., 2020*; *Shirai et al., 2016*). We therefore assessed what conformation PKM2 was expressed by cultured NK cells. This was achieved using disuccinimidylsuberate (DSS) crosslinking in unstimulated or IL-2/12-stimulated NK cells, whereby proteins in close proximity are 'linked' to each other by DSS.

The cells can then be lysed and assessed by immunoblot for PKM2 expression. Western blotting revealed that IL-2/12-stimulated NK cells express both monomeric and tetrameric PKM2 (*Figure 1e*). Therefore, PKM2 expression is robustly increased in activated NK cells, is the dominant pyruvate kinase isoform in these metabolically active cells and is present as both monomers and tetramers.

## PKM2<sup>NK-KO</sup> mice show no defects in splenic NK cell development and function

To investigate the importance of PKM2 during NK cell responses, NK-cell-specific *Pkm2* knockout mice were generated by backcrossing mice with loxP sites flanking the exon specific for *Pkm2*, exon 10, with mice expressing Cre recombinase under the control of the *Ncr1* promoter (*Narni-Mancinelli et al., 2011*; *Israelsen et al., 2013*). NK cells were purified by cell sorting from the spleens of *Pkm2*<sup>flox/flox</sup> *Ncr1*-Cre mice (hereafter called *Pkm2*<sup>NK-KO</sup>) and control mice (*Pkm2*<sup>WT/WT</sup> *Ncr1*-Cre, hereafter called *Pkm2*<sup>NK-WT</sup>) and DNA was isolated. The *Pkm2*<sup>WT/WT</sup> or *Pkm2*<sup>fl/fl</sup> genes were amplified using PCR and electrophoresed on a DNA agarose gel (*Figure 2a*). The data show that NK cells containing both the *Ncr1*-Cre transgene and homozygous for the *Pkm2*<sup>fl</sup> locus specifically excise the exon 10 of *Pkm2* gene leading to a smaller DNA band (~200 kb) (*Figure 2a*). Remaining splenocytes (not including NK cells) show a normal sized band for *Pkm2*<sup>fl/fl</sup> (~600 kb). NK cells developed normally in *Pkm2*<sup>NK-KO</sup> mice as the numbers and frequencies of NK cells and NK cells subsets were normal in the spleen when compared to control *Pkm2*<sup>NK-WT</sup> mice (*Figure 2b,c*). Consistent with normal NK cell development splenic *Pkm2*<sup>NK-KO</sup> NK cells expanded normally ex vivo in response to low-dose IL-15, important for NK cells homeostatic proliferation (*Figure 2d*). Cultured PKM2<sup>NK-KO</sup> and PKM2<sup>NK-WT</sup> NK cells were stimulated with IL-2/IL-12 cytokine for 18 hr and analysed by flow cytometry for the expression of effector molecules. *Pkm2*<sup>NK-KO</sup> NK cells expressed comparable levels of the cytotoxic granule component granzyme B and produced equivalent amounts of IFNγ compared to *Pkm2*<sup>NK-WT</sup> NK cells (*Figure 2e–h*). Additionally, there were no differences in the secretion of cytokines (IFNγ, TNFα and IL-10) or chemokines (MIP1α and MIP1β) between stimulated *Pkm2*<sup>NK-KO</sup> and *Pkm2*<sup>NK-WT</sup> NK cells (*Figure 2i,j*).

## PKM2<sup>NK-KO</sup> mice respond normally to MCMV infection

As cellular metabolism is considered to be of particular importance in highly proliferative cells, we next investigated whether *Pkm2*<sup>NK-KO</sup> mice could control MCMV infection normally. NK cells are particularly important in the early immune response to this viral infection and a subset of Ly49H<sup>+</sup> NK cells undergo robust proliferation in MCMV infected mice. Therefore, *Pkm2*<sup>NK-KO</sup> and *Pkm2*<sup>NK-WT</sup> mice were infected with MCMV ($1 \times 10^5$ PFU from salivary passage) i.p. and analyzed 4 days post-infection. The spleen size as determined by weight was equivalent in *Pkm2*<sup>NK-KO</sup> and *Pkm2*<sup>NK-WT</sup> mice as were the numbers of NK cells and the frequencies of Ly49H<sup>+</sup> splenic NK cells (*Figure 3a–c*). In addition, there was no difference in NK cell activation in *Pkm2*<sup>NK-KO</sup> and *Pkm2*<sup>NK-WT</sup> mice based on CD69 expression. Additionally, Ly49H<sup>+</sup> cells from these mice showed equivalent levels of virus-induced proliferation based on BrDU incorporation (*Figure 3d–g*). There were no differences in the levels of IFNγ, TNFα and IL10 in the serum of uninfected or MCMV infected *Pkm2*<sup>NK-KO</sup> and *Pkm2*<sup>NK-WT</sup> mice (*Figure 3h*). Finally, viral loads on day 4 were similar in *Pkm2*<sup>NK-KO</sup> and *Pkm2*<sup>NK-WT</sup> mice (*Figure 3i*). These data clearly show that the immune response to MCMV infection was not compromised in mice containing *Pkm2* deficient NK cells 4 days post-infection.

## PKM2<sup>NK-KO</sup> NK cells adjust PKM1 expression to normalise total pyruvate kinase activity

Considering the important roles described for PKM2 in other immune cells, the lack of any functional phenotype in *Pkm2*<sup>NK-KO</sup> was intriguing. Next, we considered whether the loss of *Pkm2* affected the metabolic pathways used by *Pkm2*<sup>NK-KO</sup> NK cells. Firstly, the flux through glycolysis and OXPHOS was assessed using the Seahorse extracellular flux analyzer. The basal and maximal rates of glycolysis and of OXPHOS were comparable in both unstimulated and IL2/IL12 stimulated *Pkm2*<sup>NK-KO</sup> compared to *Pkm2*<sup>NK-WT</sup> NK cells (*Figure 4a–e*). In addition, the levels of glycolytic intermediates, pentose phosphate pathway components (PPP) and elements of the tricarboxcylic acid (TCA) cycle were similar in *Pkm2*<sup>NK-KO</sup> and *Pkm2*<sup>NK-WT</sup> NK cells (*Figure 4f–h*). Therefore, despite lacking the most highly expressed pyruvate kinase subunit (*Figure 1e*), which is known to play a crucial role in

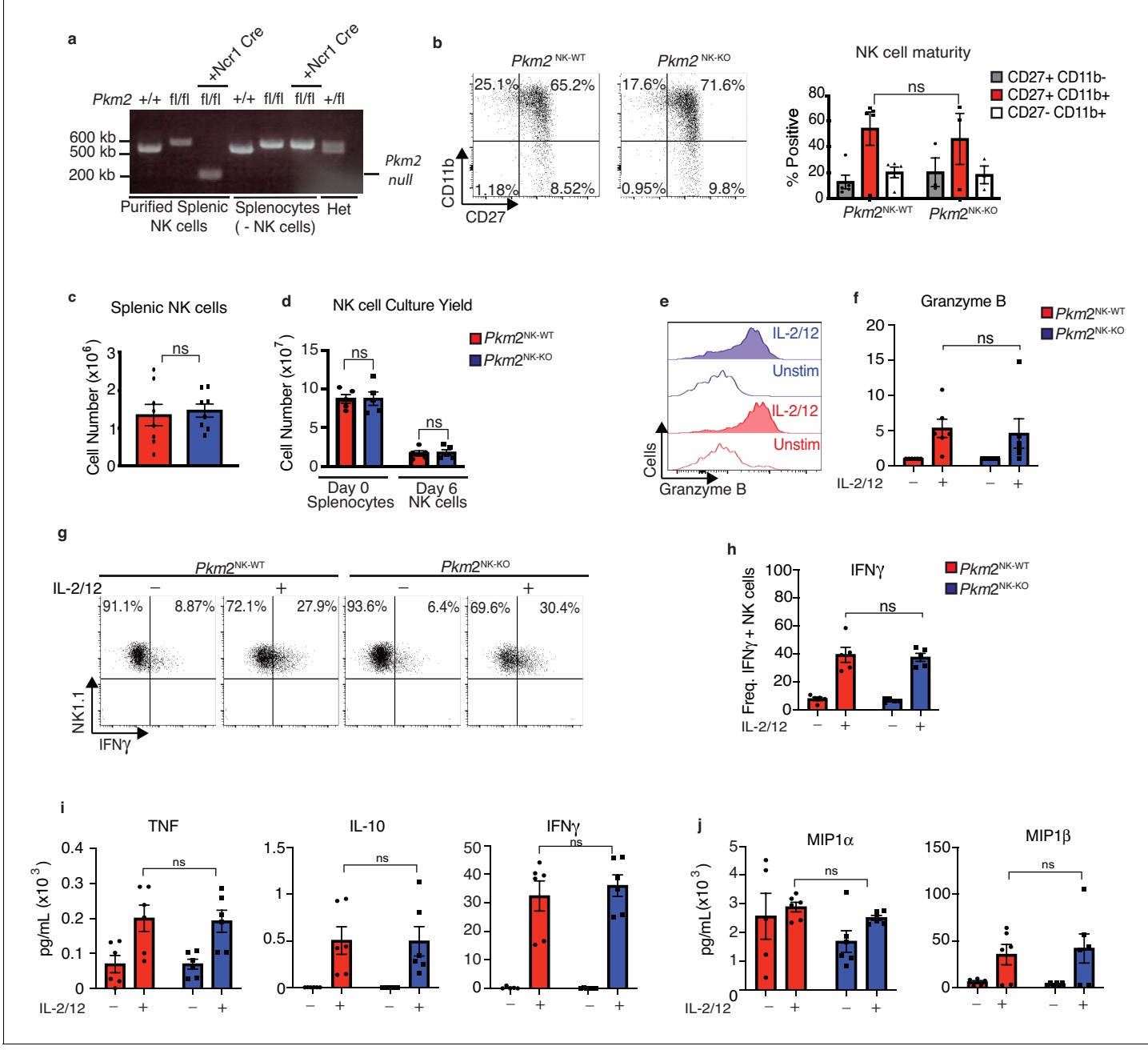

**Figure 2.** PKM2 is not required for IL-2/12 induced NK cell effector function in vitro. (a) NK cells were sorted by flow cytometry (NK1.1+NKp46+CD3−CD49b+) from wildtype C57Bl/6, *Pkm2*^fl/fl or *Ncr1*^Cre*Pkm2*^fl/fl mice. Cells were lysed and DNA was purified. DNA was subject to PCR amplification for the *Pkm2* gene and products were electrophoresed on a 1.8% agarose gel and imaged. (b) Splenic *Pkm2*^NK-WT and PKM2^NK-KO NK cells were analysed by flow cytometry for the expression of CD11b and CD27. (c) *Pkm2*^NK-WT and *Pkm2*^NK-KO cells were isolated and counted and analysed by flow cytometry for frequency of NK1.1+NKp46+CD3−. cells. (d) Splenic *Pkm2*^NK-WT and *Pkm2*^NK-WT cells were expanded for 6 days in IL-15 (15 ng/mL). Data displayed show total splenocyte numbers before expansion (pre) and pure NK cell numbers after magnetic purification (pure). (e–h) *Pkm2*^WT and *Pkm2*^KO NK cells were stimulated for 18 hr in IL-2/12 or left unstimulated. NK1.1+NKp46+CD3− cells were analysed for granzyme B (e–f) or IFNγ expression (g–h). (i–j) *Pkm2*^NK-WT and *Pkm2*^NK-KO cells were stimulated for 18 hr in IL-2/12 or left unstimulated and media supernatants were collected. Supernatants were then analysed for by cytometric bead array analysis for (i) IFNγ, IL-10, TNF, (j) MIP1α, MIP1β. (b–j) data are mean +/- S.E.M for n = 4–8 mice per group. (c) Data was analysed using a Students t test. (b, d-i) Data were analysed by two-way ANOVA with multiple comparisons. *p>0.05, **p>0.01, ***p>0.001.

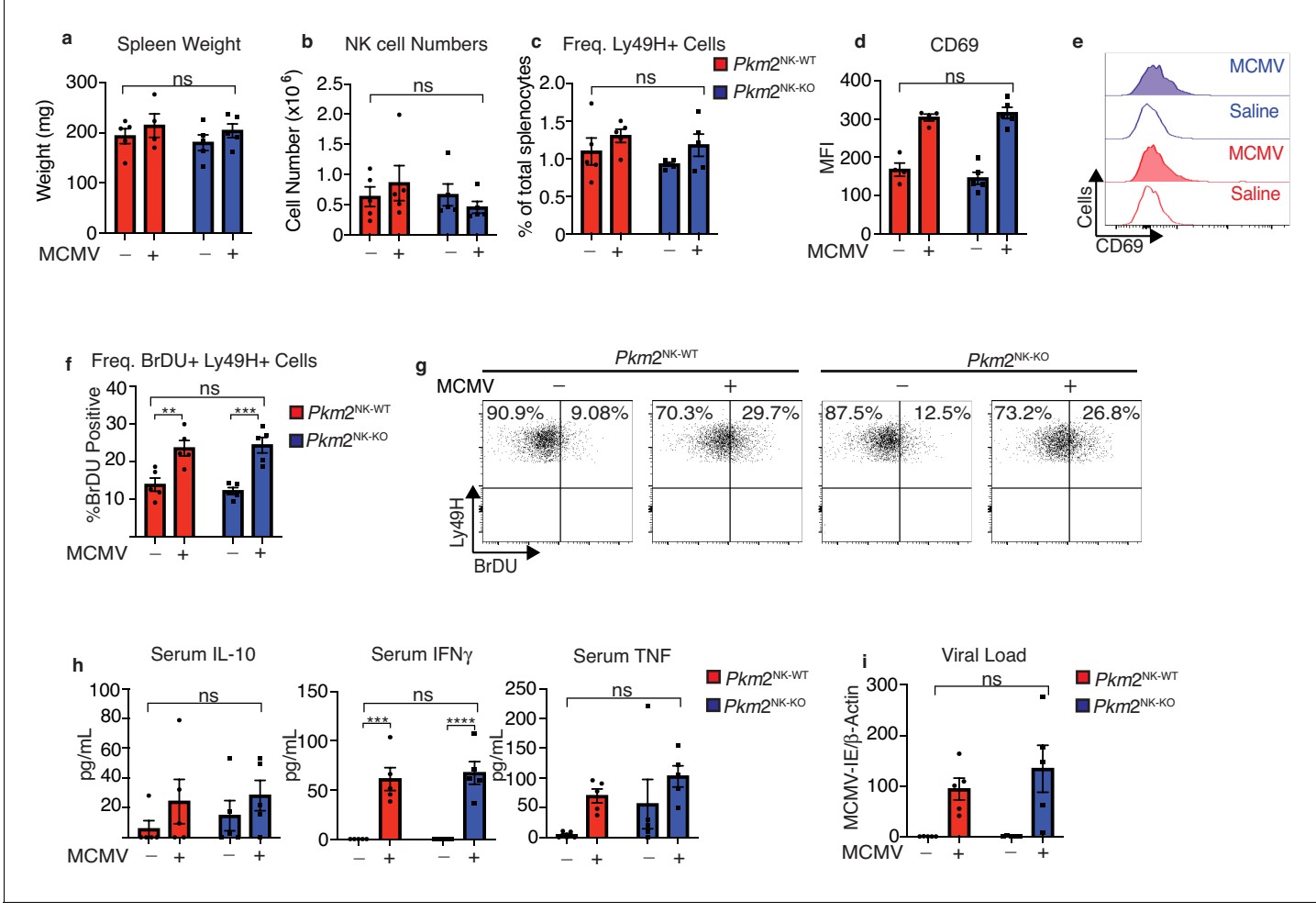

**Figure 3.** PKM2 is not required for early NK cell responses to MCMV. *Pkm2*[NK-WT] and *Pkm2*[NK-KO] mice were infected with $1 \times 10^5$ PFU of MCMV or injected with saline for 4 days. (**a**) Spleens were harvested from *Pkm2*[NK-WT] and *Pkm2*[NK-KO] mice 4 days post MCMV infection and weighed. (**b**) Spleens were harvested 4 days post-MCMV infection from *Pkm2*[NK-WT] and *Pkm2*[NK-KO] mice and NK cells were identified as being NK1.1[+]NKp46[+]CD3[-] by flow cytometry. (**c**) Ly49H-positive cells were assessed by flow cytometry and expressed as a percentage of total splenocytes (**d-e**) Splenic NK cells from *Pkm2*[NK-WT] and *Pkm2*[NK-KO] mice were identified post MCMV infection and assessed for CD69 expression by flow cytometry (**f**) Ly49H[+] cells from MCMV infected *Pkm2*[NK-WT] and *Pkm2*[NK-KO] mice were assessed for BrDU incorporation 4 days post infection (**g**) Representative dot plot of BrDU incorporation into Ly49H[+] cells 4 days post-MCMV infection. (**h**) Blood was drawn from *Pkm2*[NK-WT] and *Pkm2*[NK-KO] by cardiac puncture 4 days post-MCMV infection and serum was isolated. Serum was then analysed for levels of cytokines by cytometric bead array for IL-10, TNF and IFNγ. (**i**) Splenic viral load was measured using qPCR for MCMV-IE and DNA was normalised to β-Actin. n = 4–5 mice per group and are representative of two independent experiments. Data were analysed by two-way ANOVA with multiple comparisons. ns – not significant [*]p>0.05, [**]p>0.01, [***]p>0.001.

controlling the metabolism of other immune cell subsets, there were no substantial differences in the metabolic status of *Pkm2*-null NK cells. One possible explanation for the surprising lack of a metabolic phenotype in *Pkm2*-null NK cells could be due to compensatory changes in the expression of another pyruvate kinase isoform. Indeed, RNA and protein analyses using rtPCR, RNAseq and western blotting revealed that expression of PKM1, the alternative splice variant of *Pkm2*, was significantly increased in *Pkm2*[NK-KO] compared to *Pkm2*[NK-WT] NK cells (**Figure 4i–k**). Remarkably, *Pkm2*[NK-KO] NK cells had precisely compensated with increased PKM1 expression to maintain an extraordinarily similar rate of overall pyruvate kinase activity, as measured using a direct biochemical enzymatic assay (**Figure 4l**). This re-calibration of pyruvate kinase activity demonstrates the metabolic plasticity of NK cells.

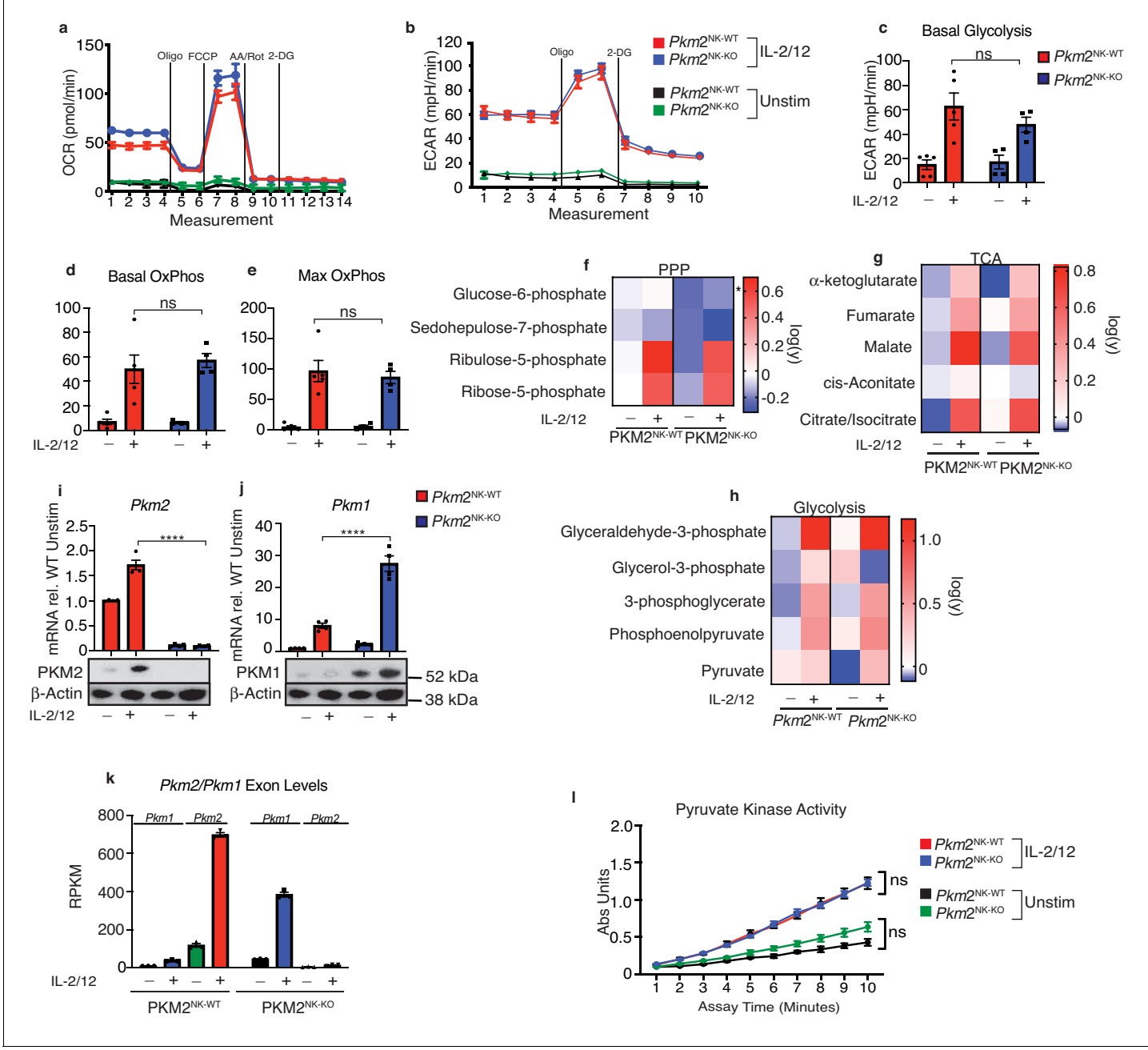

**Figure 4.** Transcriptional regulation of PKM1 can metabolically compensate for loss of PKM2. (a) *Pkm2*^NK-WT and *Pkm2*^NK-KO splenocytes were cultured for 6 days in low-dose IL-15 and NK cells were magnetically purified. NK cells were then stimulated for 18 hr in IL-2/12 or left unstimulated. (a–e) Stimulated or unstimulated *Pkm2*^NK-WT and *Pkm2*^NK-KO cells were analysed by seahorse for glycolysis and oxphos (a) *Pkm2*^NK-WT and *Pkm2*^NK-KO cells were stimulated with (IL-2/12) or left unstimulated (low-dose IL-15) and oxygen consumption was measured over time. Data is representative OCR trace. (b) Bar graph of pooled data for basal rates of OxPhos. (c) Bar graph of maximum rates of OxPhos in *Pkm2*^NK-WT and *Pkm2*^NK-KO treated with IL-2/12 or left unstimulated (d) *Pkm2*^NK-WT and *Pkm2*^NK-KO cells were stimulated with (IL-2/12) or left unstimulated (low-dose IL-15) and extracellular acidification was measured over time. Data is representative ECAR trace. (f–h) *Pkm2*^NK-WT and *Pkm2*^NK-KO cells were stimulated with (IL-2/12) or left unstimulated (low-dose IL-15) and cells were analysed for relative metabolite abundance using LC-MS metabolomics. Peak areas were normalised to the average of *Pkm2*^NK-WT unstimulated samples and then log(y) transformed using Graphpad Prism. (f) Data displayed are metabolites of the pentose phosphate pathway determined using LC-MS metabolomics. (g) Data displayed are a heat map for relative abundance of tricarboxylic cycle metabolites determined using LC-MS metabolomics. (h) Data displayed are a heat map for relative abundance of glycolytic metabolites determined using LC-MS metabolomics. (i) *Pkm2*^NK-WT and *Pkm2*^NK-KO cells were stimulated with (IL-2/12) or left unstimulated (low-dose IL-15) and cells were analysed by immunoblot or qPCR for the expression of PKM1 or PKM2. The same western blot was stripped and re-probed for PKM2 and the same loading control β-actin is pictured for both. qPCR data was normalised using the ΔΔCt method and HPRT housekeeping gene was used. (k) RNA sequencing for the

*Figure 4 continued on next page*

*Figure 4 continued*

quantity of transcripts encoding *Pkm1* and *Pkm2*. (m) *Pkm2*[NK-WT] and *Pkm2*[NK-KO] cells were stimulated with (IL-2/12) or left unstimulated (low-dose IL-15) and cells were lysed and assessed using an absorbance based assay for total pyruvate kinase activity. n = 3–5 mice per group. Data are mean +/- S.E.M and were analysed by (a–h) two-way ANOVA with multiple comparisons or (i–j) one-way ANOVA with Tukey post-test. ns – not significant *p>0.05, **p>0.01, ***p>0.001.

## PKM2 does not substantially regulate gene expression in activated NK cells

While PKM1 expression can compensate for the lack of PKM2 in terms of catalyzing the final step of glycolysis, PKM1 has not been demonstrated to have the non-glycolytic roles that have been ascribed to mono/dimeric PKM2, such as the regulation of transcription factors such as HIF1α and STAT5. Therefore, we used RNAseq analysis to investigate whether there were any differences in gene expression in *Pkm2*[NK-KO] versus *Pkm2*[NK-WT] NK cells. There were very few significant differentially expressed genes (DEG) in the transcriptome of *Pkm2*[NK-KO] versus *Pkm2*[NK-WT] NK cells (*Figure 5a*). There were only two genes upregulated by over twofold and four genes downregulated by greater than twofold expression with a false discovery rate of 0.05 (*Figure 5a*, *Supplementary file 1*). PKM2 has been linked to the control of gene expression through its regulation of HIF1α and STAT5 transcription factors (*Angiari et al., 2020*). Importantly, there was no enrichment for HIF1α or STAT5 target genes in the differentially expressed mRNA in *Pkm2*[NK-KO] NK cells (*Figure 5b and c*, *Supplementary file 2*). To further investigate the effect of PKM2 in NK cell gene expression, an acute pharmacological approach was used to manipulate PKM2 function. TEPP-46 is a pharmacological activator of PKM2 function that promotes the oligomerisation of mono/dimeric PKM2 into tetrameric, catalytically active PKM2 (*Anastasiou et al., 2012*), resulting in the loss of non-glycolytic PKM2 signalling functions (*Angiari et al., 2020*). Therefore, cultured NK cells were stimulated with IL-2/IL-12 in the presence or absence of TEPP-46 and analysed by RNA-seq. The transcriptomes were remarkably similar between the two conditions with only 10 DEG with a fold change cutoff of 2 and a FDR of 0.05 (*Figure 5d*). Amongst these genes, there was again no enrichment for HIF1α or STAT5 targets aside from a small but significant increase in *Bcl2* expression (*Figure 5e,f*). Therefore, using both genetic and pharmacological approaches these data clearly show that PKM2 is not necessary for the regulation of the NK cell transcriptome.

## Monomeric PKM2 is important for cytokine-induced NK cells responses

PKM2 tetramerisation in response to TEPP-46 results in increased PKM2 enzymatic activity leading to an increased rate of phosphoenolpyruvate conversion to pyruvate at the end of glycolysis. As expected, addition of TEPP-46 for 1 hr to cultured NK cells that had been previously stimulated for 18 hr with IL-2/IL-12, resulted in an increase in pyruvate kinase activity and of cellular glycolysis (*Figure 6a–c*). These data confirm that a substantial portion of PKM2 is present in the less catalytically active mono/dimeric conformation in IL-2/IL-12 stimulated NK cells, as its enzymatic activity can be boosted by TEPP-46. To investigate the importance of mono/dimeric PKM2 during NK cell activation, cultured NK cells were stimulated with IL-2/IL-12 for 18 hr in the presence or absence of TEPP-46. NK cells treated with TEPP-46 were viable but showed reduced production of IFNγ and reduced granzyme B expression (*Figure 6d–f*). TEPP-46 treated NK cells also secreted reduced amounts of TNFα and IL10 while maintaining normal production of MIP1α/β chemokines (*Figure 6g–h*). We confirmed that TEPP-46 was specific for PKM2 by treating *Pkm2*[NK-WT] and *Pkm2*[NK-KO] NK cells with TEPP-46 and assessing IFNγ and granzyme B production. TEPP-46 impaired *Pkm2*[NK-WT] IFNγ and granzyme B production while having no effect in *Pkm2*[NK-KO] cells. Similarly, TEPP-46 did not affect pyruvate kinase activity in *Pkm2*[NK-KO] cells (*Figure 6—figure supplement 1*).

Flow cytometric analysis revealed that NK cells stimulated in the presence of TEPP-46 were smaller in size than vehicle-treated NK cells suggesting defects in cellular growth (*Figure 7a*). TEPP-46-treated NK cells also showed reduced proliferation in response to IL2/IL12 stimulation (*Figure 7—figure supplement 1*). This was consistent with the concept that monomeric PKM2 promotes cellular growth by limiting the last step of glycolysis and allowing glycolytic intermediates to be diverted into biosynthetic pathways. In this scenario, TEPP-46 would drain glycolysis of glycolytic intermediates and reduce biosynthesis and cellular growth. However, NK cells stimulated with IL-2/IL-12 plus

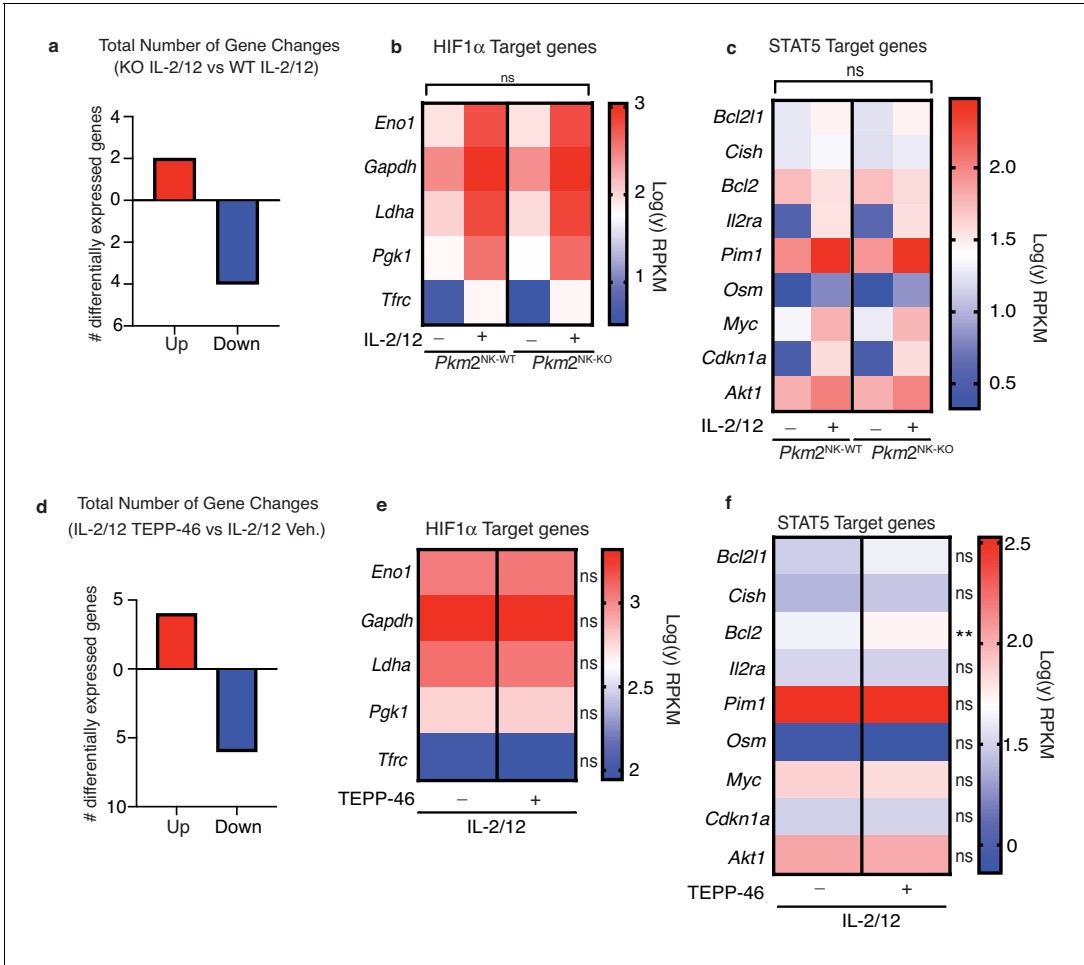

**Figure 5.** PKM2 is not required for transcription of HIF1α and STAT5α target genes in NK cells. (a–c) $Pkm2^{NK-WT}$ and $Pkm2^{NK-KO}$ cultured and purified cells were stimulated with (IL-2/12) or left unstimulated (low-dose IL-15) for 18 hr. HiSeq RNA sequencing was then performed. (a) Differential gene expression analysis was carried out to assess total gene changes between IL-2/12 stimulated $Pkm2^{NK-WT}$ and $Pkm2^{NK-KO}$ cells. Total gene changes were assessed at a fold change cut-off of 2 and a p value of 0.05 with an FDR of 0.05. (c) Expression levels of key HIF1α target genes were assessed and compared between $Pkm2^{NK-WT}$ and $Pkm2^{NK-KO}$ cells (c) Expression levels of key STAT5 target genes were assessed and compared between $Pkm2^{NK-WT}$ and $Pkm2^{NK-KO}$ cells. (d-f) Cultured wildtype NK cells were stimulated with IL-2/12 +/- TEPP-46/vehicle for 18 hr. HiSeq RNA sequencing was then performed. (d) Differential gene expression analysis was carried out to assess total gene changes between IL-2/12 TEPP-46 (50 μM) and IL-2/12 Vehicle (0.1% v/v DMSO). Total gene changes were assessed at a fold change cut-off of 2 and a p value of 0.05 with FDR 0.05. (e) Expression levels of key HIF1α target genes were assessed and compared between IL-2/12 TEPP-46 (50 μM) and IL-2/12 Vehicle (0.1% v/v DMSO) treated cells. (f) Expression levels of key STAT5 target genes were assessed and compared between IL-2/12 TEPP-46 (50 μM) and IL-2/12 Vehicle (0.1% v/v DMSO) treated cells. Data are from n = 3 biological replicates per group and are displayed as mean values. (b–c) RPKM values were normalised to the average of PKM2$^{NK-WT}$ unstimulated. Fold change values were then log transformed and displayed in heat maps. (e–f) RPKM values were normalised to the mean of both groups combined and then converted to fold change from the mean. Data were then log transformed and displayed as heat maps. n = 3 mice per group. Data are mean and were analysed by (a–c) two-way ANOVA with multiple comparisons or (d-f) one-way ANOVA with Tukey post-test. ns – not significant *p>0.05.

TEPP-46, the basal rates of glycolysis were equivalent to those of NK cells stimulated with IL-2/IL-12 alone (*Figure 7b*). There was also a small but significant effect on NK cell glycolytic capacity (*Figure 7c*). While metabolomics analysis showed no decrease in the levels glycolytic intermediates there were decreased levels of the PPP metabolite ribose-5-phosphate (R5P) arguing that in the presence of TEPP-46 glucose-6-phosphate might be withheld from the PPP (*Figure 7d,e*). Since the PPP is important for the biosynthesis of nucleotides and for maintaining redox balance in the cell, we considered whether decreased flux through the PPP in response to TEPP-46 might lead to increased cellular ROS levels. Indeed, TEPP-46-treated NK cells showed a dramatic increase in ROS compared to vehicle controls, as measured using the flow cytometric ROS probe H2-DCFDA

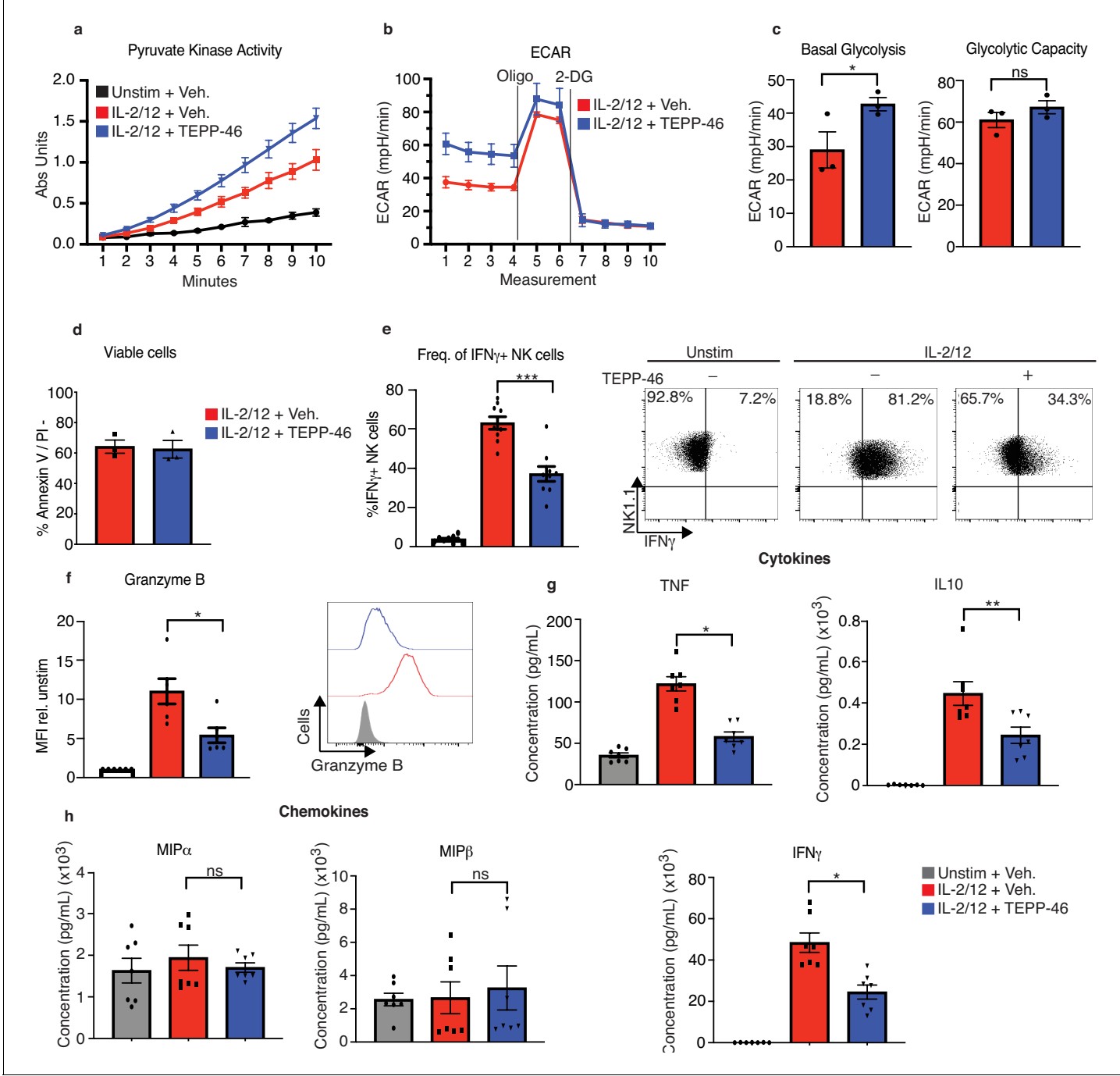

**Figure 6.** TEPP-46 activation of PKM2 is inhibitory to NK cell proinflammatory cytokine production. (**a**) Wildtype cultured NK cells were stimulated for 17 hr in IL-2/12 or left unstimulated. NK cells were then treated with TEPP-46 for 1 hr, lysed and a pyruvate kinase activity assay was carried out. (**b**) Wildtype cultured NK cells were stimulated for 17 hr in IL-2/12 or left unstimulated. NK cells were then treated with TEPP-46 for 1 hr and glycolysis was assessed by Seahorse extracellular flux analysis. (**c**) Pooled data for basal glycolysis and glycolytic capacity as measured by seahorse extracellular flux analysis. (**d**) Cultured NK cells were stimulated with IL-2/12 for 18 hr +/- TEPP-46 (50 μM) and analysed by flow cytometry for Annexin V staining and PI incorporation. (**e**) Cultured NK cells were stimulated with IL-2/12 for 18 hr +/- TEPP-46 (50 μM) or left unstimulated and analysed by flow cytometry for frequency of IFNγ expression. (**f**) Cultured NK cells were stimulated with IL-2/12 for 18 hr +/- TEPP-46 (50 μM) or left unstimulated and analysed by flow cytometry for granzyme B expression. (**g**) Cultured NK cells were stimulated with IL-2/12 for 18 hr +/- TEPP-46 (50 μM) or left unstimulated. Supernatants were harvested and analysed for levels of proinflammatory cytokines (TNF, IFNγ, IL-10) by flow cytometric bead array. (**h**) Cultured NK cells were stimulated with IL-2/12 for 18 hr +/- TEPP-46 (50 μM) or left unstimulated. Supernatants were harvested and analysed for levels of chemokines (MIP1α and MIP1β) by flow cytometric bead array. (**a–d**) data are representative of three independent experiments. (**e**) Data are pooled data of 9

*Figure 6 continued on next page*

Figure 6 continued

experiments. (f) Data are pooled data of six experiments. (g–h) Data are pooled data of seven individual experiments. Data are mean +/- S.E.M and were analysed by (a–c) two-way ANOVA with multiple comparisons or (d-f) one-way ANOVA with Tukey post-test. ns – not significant *p>0.05, **p>0.01, ***p>0.001.

The online version of this article includes the following figure supplement(s) for figure 6:

**Figure supplement 1.** TEPP-46 does not affect IFNγ or granzyme B production in *Pkm2*[NK-KO] cells.

(*Figure 8a*). Inhibition of flux through the PPP using the glucose-6-phosphate dehydrogenase inhibitor, DHEA, similarly resulted in an increase in ROS (*Figure 8b*). The PPP supports redox balance because it generates NADPH, the electron donor required for the redox enzymes glutathione reductase and thioredoxin reductase. Accordingly, we found a significant decrease in the levels of NADPH in TEPP-46-treated NK cells (*Figure 8c*). Moreover, supporting that there was increased oxidative

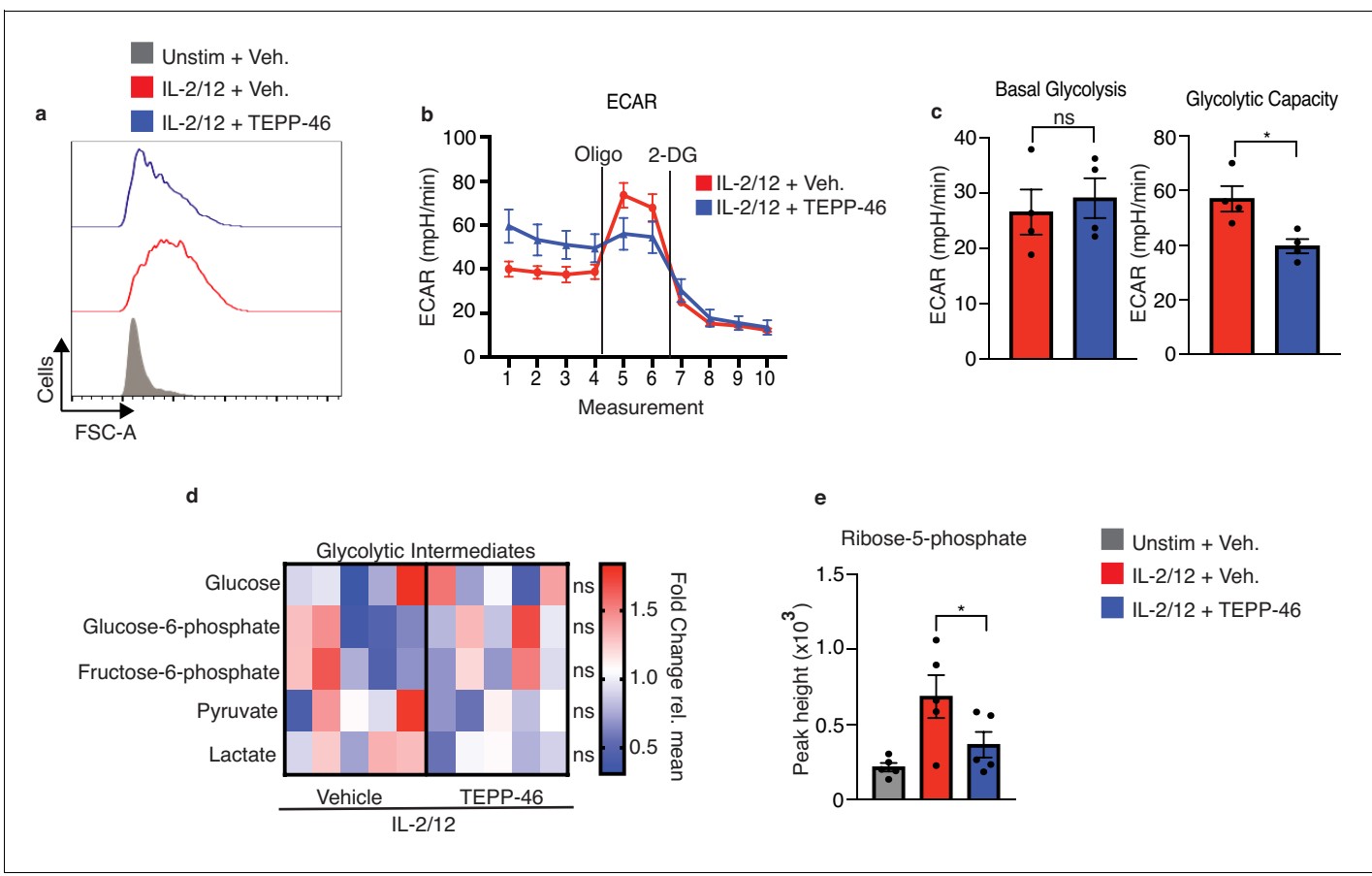

**Figure 7.** PKM2 activation inhibits normal cell growth and pentose phosphate pathway fuelling. (a) Cultured NK cells were stimulated with IL-2/12 for 18 hr +/- TEPP-46 (50 μM) or left unstimulated. NK cells were analysed by flow cytometry and forward scatter (FSC-A) was assessed (b–c) Cultured NK cells were stimulated with IL-2/12 for 18 hr +/- TEPP-46 (50 μM) and analysed for glycolysis by Seahorse extracellular flux analysis (b) representative seahorse trace for IL-2/12 stimulated NK cells +/- TEPP-46 (18 hr) (c) IL-2/12 stimulated NKs treated +/- TEPP-46 (18 hr) were analysed by Seahorse extracellular flux analysis and data were compiled for glycolytic capacity and pooled basal glycolysis. (d) Cultured NK cells were stimulated with IL-2/12 for 18 hr +/- TEPP-46 (50 μM) and analysed by metabolomics for glycolytic metabolites using GC-MS metabolomics. Data were normalised to the mean of each metabolite peak height across both groups and displayed as fold change relative to the mean. (e) Metabolomics analysis for the metabolite ribose-5-phosphate displayed as peak height. Data are pooled or representative of between three and five individual experiments. Data were analysed by a Students t test (b) or by one-way ANOVA with Tukey post-test (e) Data are representative of mean +\- S.E.M. ns – not significant *p>0.05, **p>0.01, ***p>0.001.

The online version of this article includes the following figure supplement(s) for figure 7:

**Figure supplement 1.** TEPP-46 treatment prevents normal cytokine-induced NK cell proliferation.

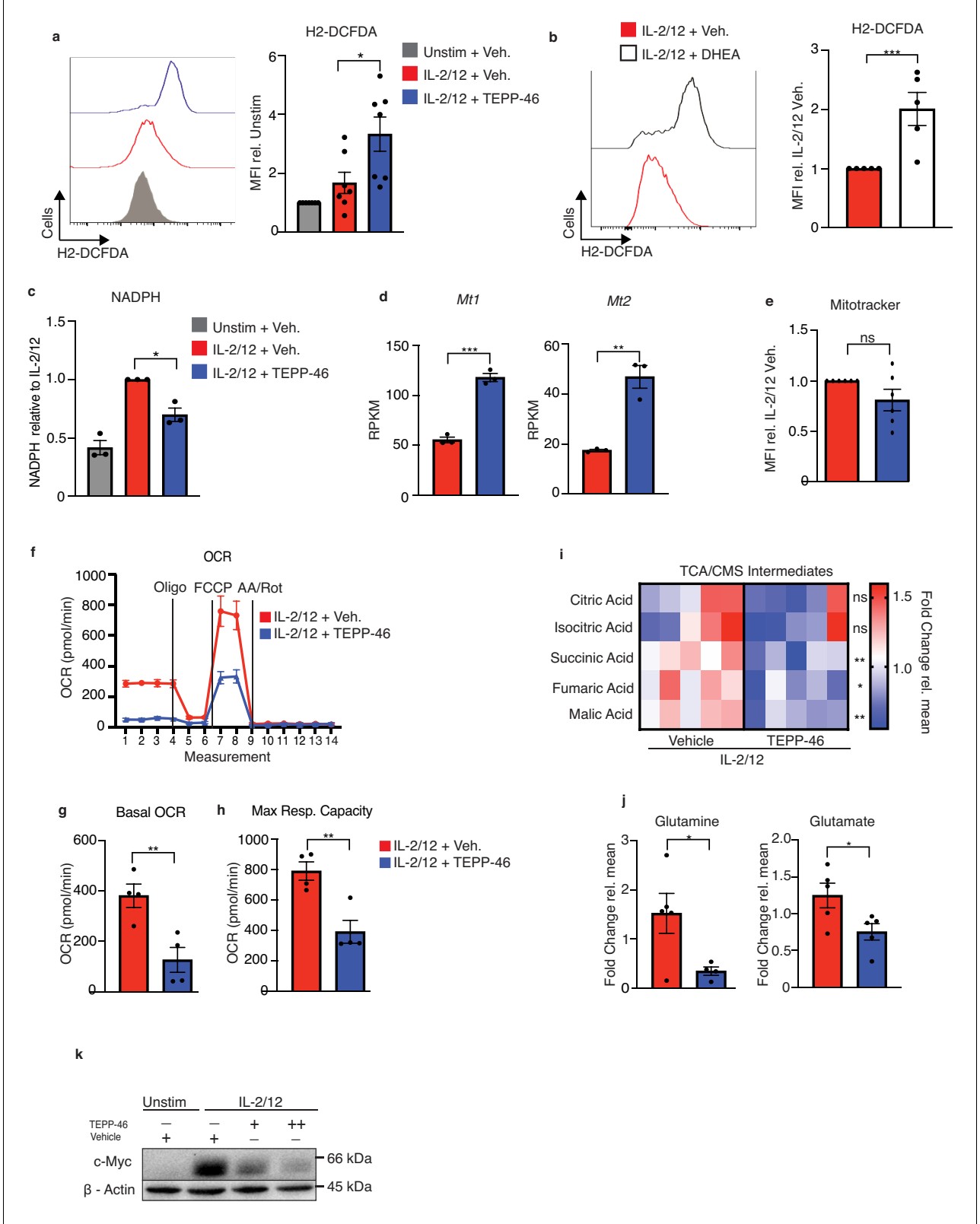

**Figure 8.** PKM2 activation inhibits normal NK cell oxidative metabolism fuelling. (a) Cultured NK cells were stimulated with IL-2/12 for 18 hr +/- TEPP-46 (50 μM) or left unstimulated (low-dose IL-15) and stained for ROS using the flow cytometric probe DCFDA. NK cells were gated (NK1.1⁺NKp46⁺CD3⁻) and MFI for DCFDA was analysed and displayed as both a representative histogram and pooled data from seven individual experiments. (b) Cultured NK cells were stimulated with IL-2/12 for 18 hr +/- DHEA (75 μM) or vehicle and stained for ROS using the flow cytometric probe DCFDA. NK cells were

*Figure 8 continued on next page*

*Figure 8 continued*

gated (NK1.1⁺NKp46⁺CD3⁻) and MFI for DCFDA was analysed and displayed as both a representative histogram and pooled data from five individual experiments. (**c**) Cultured NK cells were stimulated with IL-2/12 for 18 hr +/- TEPP-46 (50 μM) or left unstimulated (low-dose IL-15). Cells were lysed and assessed for NADPH levels using a luminescent NADPH assay with $0.15 \times 10^6$ cells with three technical replicates per assay. Data are pooled data of three independent experiments. (**d**) Cultured wild-type NK cells were stimulated with IL-2/12 +/- TEPP-46/vehicle (0.1% DMSO) for 18 hr. HiSeq RNA sequencing was then performed. Expression levels for the genes *Mt1* and *Mt2* are displayed as RPKM (Reads Per Kilobase of transcript, per Million mapped reads) and are pooled data from three individual experiments. (**e**) Cultured NK cells were stimulated with IL-2/12 for 18 hr +/- TEPP-46 (50 μM) or vehicle and using the flow cytometric probe Mitotracker red. NK cells were gated (NK1.1⁺NKp46⁺CD3⁻) and MFI for mitotracker was analysed and displayed as both a representative histogram and relative pooled data from six individual experiments. (**f–h**) Cultured NK cells were stimulated with IL-2/12 for 18 hr +/- TEPP-46 (50 μM) and analysed for oxygen consumption by Seahorse extracellular flux analysis (**f**) representative seahorse trace for IL-2/12 stimulated NK cells +/- TEPP-46 (18 hr). (**g–h**) IL-2/12 stimulated NKs treated +/- TEPP-46 (18 hr) were analysed by Seahorse extracellular flux analysis and data were compiled for basal OCR and maximum respiratory capacity. (**i**) Cultured NK cells were stimulated with IL-2/12 for 18 hr +/- TEPP-46 (50 μM) and analysed by metabolomics for tricarboxylic acid/citrate-malate shuttle metabolites using GC-MS metabolomics. Data were normalised to the mean of each metabolite peak height across both groups and displayed as fold change relative to the mean. (**j**) Metabolomics analysis for the metabolite glutamine and glutamate displayed as peak height. One outlier was omitted from glutamine using a Grubbs test (α = 0.05). (**k**) Cultured NK cells were left unstimulated or activated with IL-2/12 +/- TEPP-46 (25 or 50 μM) for 18 hr were lysed for protein. Samples were analysed by immunoblot for cMyc and β-Actin protein expression. Data are pooled or representative of between three and five experiments. Data are representative of mean + \- S.E.M. (**a,c**) Data were analysed by one-way ANOVA with Tukey post-test. (**b,d–e,g–j**) Data were analysed using students t test ns – not significant *p>0.05, **p>0.01, ***p>0.001.

stress in TEPP-46-treated NK cells, our RNAseq data showed that two of the most significantly upregulated genes in TEPP-46-treated NK cells were members of the metallothionein family, namely *Mt1* and *Mt2* (*Figure 8d*). Metallothioneins are a set of zinc-responsive proteins that have antioxidant properties and are known to be induced in response to oxidative stress (*Ruttkay-Nedecky et al., 2013*).

As oxidative stress can be linked to mitochondrial damage, we next investigated mitochondrial function in NK cells stimulated with or without TEPP-46. While there was no difference in the membrane potential-dependent mitochondrial mass in TEPP-46 treated NK cells (*Figure 8e*), we found that there was a substantial decrease in the basal rates of OXPHOS (*Figure 8f*). Additionally, there was a greatly decreased maximum respiration and spare respiratory capacity (*Figure 8g,h*). Normal mitochondrial mass with reduced maximum respiration suggest that there was reduced fueling of OXPHOS. Two prominent metabolic cycles that feed electrons into the mitochondrial electron transport chain are the TCA cycle and the Citrate-Malate shuttle (*Assmann et al., 2017*; *Loftus et al., 2018*). The metabolites of both these cycles were reduced in TEPP-46 treated NK cells (*Figure 8i*). While malate is shared between both these metabolic cycles, succinic and fumaric acids are specific for the TCA cycle. In line with decreased fueling of oxidative metabolism, the levels of glutamine and glutamate were also reduced indicative of impaired glutaminolysis (*Figure 8j*). The transcription factor cMyc is important for glutamine metabolism and was also found to be reduced in TEPP-46-treated NK cells; however, this was only at the protein level as it was not found to be altered in the transcriptomic dataset (*Figure 8k*). Considering the increased ROS in TEPP-46-treated NK cells, we investigated whether the use of cellular antioxidants could reverse NK cell dysfunction. Interestingly, the addition of antioxidant MitoQ was sufficient to partially restore the expression of cMyc protein expression in TEPP-46-treated NK cells (*Figure 9a*). We could also profoundly reduce TEPP-46 induced ROS using the antioxidant N-acetylcysteine (NAC) (*Figure 9b*). In addition, NAC treatment could rescue the production of IFNγ, but not that of TNF in NK cells stimulated with IL2/12+TEPP46 (*Figure 9c*).

Taken together, this study demonstrates that PKM2 is involved in metabolic but not transcriptional regulation of NK cell responses.

## Discussion

PKM2 is a metabolic enzyme that has received significant attention in the immunometabolism field because in addition to its role in glycolysis it can have non-metabolic functions, augmenting transcriptional programmes to regulate immune function. This 'moonlighting' activity has been demonstrated in macrophages, T cells and cancer cells where PKM2 supports the function of transcription factors including HIF1α and STATs (*Angiari et al., 2020*; *Palsson-McDermott et al., 2015*;

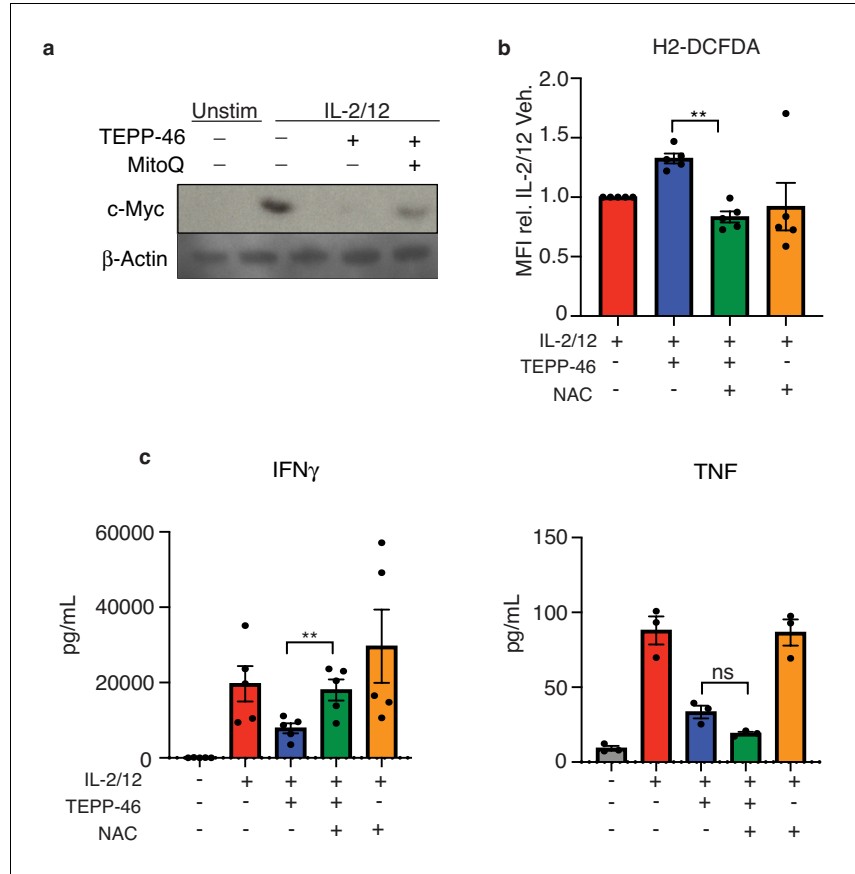

**Figure 9.** Antioxidant treatment can protect against TEPP-46 induced ROS. (**a**) Cultured NK cells were stimulated with IL-2/12 for 18 hr +/- TEPP-46 (50 μM) in the presence or absence of MitoQ (50 nM) or left unstimulated (low-dose IL-15). Cells were then lysed and analysed by immunoblot for cMyc and β-Actin expression. (**b**) Cultured NK cells were stimulated with IL-2/12 for 18 hr +/- TEPP-46 (50 μM) and NAC (7.5 mM) or vehicle and stained for ROS using the flow cytometric probe DCFDA. NK cells were gated (NK1.1[+]NKp46[+]CD3[-]) and relative MFI for DCFDA was analysed and displayed pooled data from five individual experiments. (**c–e**) Cultured NK cells were stimulated with IL-2/12 for 18 hr +/- TEPP-46 (50 μM) and NAC (7.5 mM) or vehicle. Supernatants were harvested and analysed for levels of cytokines (TNF, IFNγ) by flow cytometric bead array. (**a**) Representative western blot of three individual experiments (**b–e**) Data are mean +/- S.E.M and are representative of three to five individual experiments. Data was analysed using one way ANOVA with multiple comparisons. [**]p>0.01, ns – non significant.

*Zhang et al., 2019*). Our study shows that PKM2 has no significant role in regulating HIF1α and STAT associated gene expression in NK cells. The data clearly show that while PKM2 is the most highly expressed isoform in NK cells, either deletion of PKM2 or allosteric tetramerisation of PKM2 has minimal impact upon the NK cell transcriptome and that there was no evidence of significantly altered expression of HIF1α or STAT target genes. This study highlights that diversity of regulatory circuits in immune cells and shows the importance of verifying mechanisms in individual immune cell subsets.

Stimulated NK cells preferentially express PKM2 over PKM1 for metabolic reasons. Unlike PKM1, PKM2 can exist in different conformations that have different catalytic activities. PKM2 tetramers are highly enzymatically active and have a Km for phosphoenolpyruvate of 0.03 mM, whereas PKM2 mono/dimers are considerably less active with a Km of 0.46 mM as measured in breast cancer cells (*Mazurek, 2011*). Therefore, the expression of PKM2 gives NK cells the ability to quickly regulate their glycolytic flux toward anabolic or catabolic processes and confers upon them a level of metabolic plasticity. It has been previously demonstrated that NK cells lacking the metabolic regulator mTORC1 are defective in responding to MCMV challenge (*Marçais et al., 2014*). Similarly, mice treated with 2-deoxyglucose, a glycolytic inhibitor are impaired in their responses to MCMV

(*Mah et al., 2017*). However, in this study, genetic deletion of *Pkm2* resulted in a normal phenotype 4 days post-MCMV infection. *Pkm2*<sup>NK-KO</sup> cells only express PKM1 which in theory should be kinetically more efficient at producing pyruvate than *Pkm2*<sup>NK-WT</sup> cells. It would have been expected that these PKM1-only expressing cells would demonstrate dysregulated metabolism and higher levels of glycolytic flux. However, the PKM1-only expressing NK cells in this study demonstrated a remarkable ability to transcriptionally titrate the overall level of PKM1 to match a similar level of glycolytic flux to *Pkm2*<sup>NK-WT</sup> cells in vitro. We propose that this compensation is sufficient to allow NK cells to mount a normal immune response to MCMV infection, a process for which regulated NK cell metabolism is known to be important.

Although, PKM1 can metabolically compensate for PKM2, there is no evidence in the literature that PKM1 can substitute for PKM2-mediated signalling and transcriptional regulation. The data presented in this study show that PKM2 deletion does not adversely affect previously described PKM2-regulated signalling axes. These data are surprising as PKM2 has been previously shown to regulate cMyc, HIF1α and STAT5 signalling in CD4 T cells (*Angiari et al., 2020*). Although cMyc was reduced with TEPP-46 treatment in this study, it is important to note that this effect is secondary PKM2 metabolic regulation of NK cells as it is TEPP46-induced ROS production that leads to reduced cMyc protein expression. This was demonstrated by the rescued cMyc expression upon the addition of antioxidants. However, the reduced levels of cMyc protein were not sufficient to have a substantial impact on the expression of described cMyc target genes as there were only a total of 6 DEG (*Supplementary file 1*). The increased ROS in TEPP46 treated cells is likely to have accumulated over the 24 hr activation period and by extension it is likely that cMyc expression was not impaired at earlier time points. This may explain the lack of a cMyc gene signature in the list of differentially expressed genes identified in the RNAseq analysis. It will be of interest to further explore the relationship between PKM2, ROS and cMyc in the regulation of NK cell function.

It is also tempting to speculate that the profound differences in PKM2 function between NK cells and other cell types may be due to both functional and technical reasons. Although NK cells and T cells are both lymphocytes and have similar functional roles, their methods and contexts of activation vastly differ. T cells require a two-signal activation composed of both TCR/costimulatory molecule ligation and/or IL-2 signalling. In the context of PKM2-mediated T cell metabolic signalling, it is possible that receptor-mediated signalling may play an important role in activating the non-metabolic functions of PKM2. Indeed, in TCR stimulated T cells, levels of phosphoenolpyruvate (a PKM substrate) positively regulates activation of nuclear factor of activated T cells (NFAT) (*Ho et al., 2015*). Therefore, in theory, increased PKM2 activity would drain the pool of phosphoenolpyruvate, resulting in decreased NFAT activation in response to TCR ligation. It is interesting to note that NFAT has been previously shown to regulate transcription at the cMyc promoter (*Mognol et al., 2012*). Therefore this may somewhat explain why NK cells have a less profound signalling phenotype than CD4+ T cells. In this study, NK cells were activated in vitro using cytokines. It is interesting to speculate that receptor mediated activation of NK cells may similarly reveal a signalling role for PKM2 in vitro.

From a technical point of view, it may also be that the method of PKM2 manipulation plays a role in whether PKM2 is required for transcription in different experimental systems. Indeed, most studies investigating PKM2 in immune cells using genetic approaches utilise shRNA to knock it down (*Kono et al., 2019*) or acute excision of exon 10 using tamoxifen Cre models in vitro (*Palsson-McDermott et al., 2015*). Knockdown approaches do not always lead to total deletion of the protein and may have differential effects to a total knock out (*Zimmer et al., 2019*). Similarly, studies that utilise a tamoxifen Cre method of PKM2 ablation may also see more acute effects of PKM2 deletion. For example, Palsson-McDermott et. al, utilised a tamoxifen Cre x *Pkm2*<sup>fl/fl</sup> mouse model and observed some interesting transcriptional effects on HIF1α target genes such as *Ldha*. This Cre model differs from the one utilised in our current study as using the *Ncr1* Cre model involves deletion of PKM2 in vivo and during the immature stage of NK cell development (*Narni-Mancinelli et al., 2011*). It is possible that deletion of PKM2 during NK cell development in vivo in this study may allow for selection for cells that have PKM2-independent transcriptional programs. It is also possible to speculate that earlier deletion of PKM2, for example, in haematopoietic progenitor stage using a *Vav* Cre model may reveal an interesting developmental phenotype (*Georgiades et al., 2002*).

The combination of genetic and pharmacological approaches is important for studying metabolic systems and while each has its limitations, they combine to provide confidence in the overall results

and conclusions. Herein, both approaches show the PKM2 does not critically regulate the NK cell transcriptome. However, both genetic and pharmacological approaches support an important metabolic role for PKM2 in NK cells. In PKM2$^{\text{NK-WT}}$ cells the recalibration of PKM1 levels indicates the importance of maintaining exact pyruvate kinase activity. Acute activation of PKM2 using TEPP-46 showed that controlling pyruvate kinase activity is important as it supports flux through the PPP to maintain redox balance in NK cells. Rerouting of glucose into the PPP has been shown to occur immediately after oxidative injury in skin cells (*Kuehne et al., 2015*) and PPP-controlled antioxidant response has also been shown to be important in the control of inflammatory macrophage responses (*Baardman et al., 2018*). Similarly CD4+ T cells from patients with rheumatoid arthritis (RA) divert glucose into the PPP to increase NADPH production for protection against ROS. This allows these RA-associated T cells to bypass normal cell cycle control points and to become hyperinflammatory (*Yang et al., 2016*). These studies indicate that there is an intricate relationship between PPP-derived antioxidants and immune function. Interestingly, PKM2 has been shown to be a key regulator of antioxidant defence in esophageal squamous cell carcinoma, through direct regulation of flux into the PPP (*Fukuda et al., 2015*).

We have demonstrated that PKM2-controlled metabolism is required for normal NK cell responses as treatment with TEPP-46 is detrimental to the production of a range of NK cell cytokines including IFNγ and TNFα. Moreover, mitochondrial damage (*Zheng et al., 2019*) or directly targeting OXPHOS using pharmacological inhibitors has also been shown to inhibit NK cell cytokine production (*Keating et al., 2016*; *Kedia-Mehta et al., 2019*) and accordingly we found that one metabolic consequence of TEPP-46 treatment is the inhibition of mitochondrial OXPHOS.

TEPP-46 has been tested as an anti-tumour drug, whereby it has been shown to inhibit tumour growth in xenograft models in *nu/nu* mice which lack a functional immune system (*Anastasiou et al., 2012*). However, it is important to consider the impact that such therapies would have for the antitumour immune responses. Although TEPP-46 showed some promise in preventing tumour growth, it is tempting to speculate that its efficacy may be hindered in mouse models with complete immune systems. As shown in this study, TEPP-46 is detrimental to NK cell metabolism and function. Similarly TEPP-46 has been shown to be anti-inflammatory in EAE models (*Angiari et al., 2020*). Therefore, PKM2 targeting therapies may not be suitable for treating cancer unless they can be specifically targeted to tumour cells.

## Materials and methods

### Mice

C57BL/6J mice were purchased from Harlan (Bicester, U.K.) and maintained in compliance with Irish Department of Health and Children regulations and with the approval of the University of Dublin's ethical review board. Mice were also obtained from NCI Frederick. Animal care at NCI Frederick was provided in accordance with procedures in: 'A Guide for the Care and Use of Laboratory Animals'. Ethics approval for the animal experiments detailed in this manuscript was received from the Institutional Animal Care and Use Committee (Permit Number: 000386) at the NCI-Frederick. *Pkm2*$^{\text{fl/fl}}$ mice were obtained from Matthew Vander Heiden at MIT, MA (*Israelsen et al., 2013*). *Ncr1*$^{\text{Cre}}$ KI mice were obtained from Eric Vivier, INSERM, France (*Narni-Mancinelli et al., 2011*). All mice were backcrossed to a C57BL/6 background. *Pkm2*$^{\text{fl/fl}}$ mice and *Ncr1*$^{\text{Cre}}$ KI mice were bred together and controls were maintained in the same room at NCI Frederick.

### Mouse genotyping

DNA samples were obtained from mice by tail snip. Tails were digested using 50 µl of tail lysis buffer and 5 µl of 10 mg/ml proteinase K in 1.5 ml tubes. Tubes were placed in a heating block at 55°C for 4 hr to overnight, allowing the tails to become digested. Tails were then heated to 95°C for 10 min to inactivate proteinase K. 950 µL of DNAse free water was then added to sample. DNA samples were stored at −20°C until use. *Pkm2*$^{\text{fl/fl}}$ genotyping was carried out using the following primers: Forward: 5'-TAG GGC AGG ACC AAA GGA TTC CCT-3', Reverse: 5'-CTG GCC CAG AGC CAC TCA CTC TTG-3'. PCR reactions were carried out using GoTaq green PCR reaction buffer (Promega). DNA was cycled according to the following protocol: 94°C for 3 min, 94°C for 30 s (x30), 59°C for 30 s (x30), 72°C for 30 s (x30). DNA was then electrophoresed on an agarose gel stained with ethidium

bromide. For *Ncr1* Cre genotyping, briefly, the following primers were utilised: Forward - 5′ GGA ACT GAA GGC AAC TCC TG- 3′, Reverse (WT)– 5′- TTC CCG GCA ACA TAA AAT AAA-3′, Reverse (Cre) - 5′ -CCC TAG GAA TGC TCG TCA AG-3′. PCR reactions were carried out using GoTaq green PCR reaction buffer (Promega). DNA was cycled according to the following protocol: 94℃ for 3 min, 94℃ for 30 s (x32), 57℃ for 30 s (x32), 72℃ for 1 min (x32), 72℃ for 3 min. DNA was then electrophoresed on an agarose gel stained with ethidium bromide.

## Cell culture

Splenocytes were isolated and cultured in IL-15 (15 ng/ml; PeproTech) at 37℃ for 6 days. On day 4, the cells were supplemented with IL-15 (15 ng/ml) and cultured for an additional 2 d. On day 6, cultured NK cells were magnetically purified (NK cell Isolation Kit II, Miltenyi) and stimulated for 18 hr with IL-2 (20 ng/ml; National Cancer Institute Preclinical Repository) and/or IL-12 (10 ng/ml; Miltenyi Biotec) cytokines. Low-dose IL-15 (6.66 ng/ml) was added as a survival factor to unstimulated cultures. Experiments were carried out in the presence or absence of TEPP-46 (Cayman Chemical or EMD Millipore), rapamycin (20 nM; Fisher), N-acetyl-cysteine (7.5 mM; EMD Millipore) or dehydroepiandrosterone (DHEA) (75 μM, Sigma Aldrich). Splenocytes were cultured in RPMI medium containing 10% FBS, 2 mM glutamine (Thermo Fisher), 50 μM 2-ME (Sigma-Aldrich), and 1% Penicillin/Streptomycin (Thermo Fisher).

## NK cell activation with polyinosinic-polycytidylic acid in vivo

Mice were injected i.p. with 100 or 200 μg polyinosinic-polycytidylic acid [poly(I:C)] in saline (InvivoGen). Mice were sacrificed after 24 hr. Spleens were harvested, and NK cells were analysed.

## Murine cytomegalovirus infection model

Stock of MCMV was a gift from the Michael Brown lab at the University of Virginia. MCMV stock was obtained from salivary gland passage of MCMV infected BALB/c mice. $1 \times 10^5$ PFU of virus in 100 μL was injected into the left side of the peritoneum in saline. Vehicle controls were given 100 μL of saline only. On day 4 post-infection, 4 hr prior to harvest, mice were injected with BrDU in saline i.p. Mice were then euthanised after 4 hr and tissues harvested. DNA from spleen was digested overnight with DNeasy extraction kit (Qiagen) for further downstream qPCR analysis.

Primers were from Integrated DNA Technologies. Primers used were as follows: Forward:5′-TG TGTGGATACGCTCTCACCTCTAT-3′, Rev:5′GTTACACCAAGCCTTTCCTGGAT-3′ (Integrated DNA Technologies). Taqman Probe was obtained from TaqMan Thermo Fisher 5′-TTCATCTGCTGCCA TACTGCCAGCTG-3′. Data were normalised to the probe for housekeeping gene β-Actin (Thermo Fisher). Primer sequences were obtained from a previously published paper by *Tanaka et al., 2007*. qPCR data was obtained using the following reaction master mix: 900 nM forward primer, 900 nM reverse primer, 200 nM Taqman probe, 1 X EagleTaq master mix (Roche) and 50 ng/ μL of DNA in 20 μL reaction volume. Master mix then underwent thermal cycling according to the following protocol: 50℃ for 2 min, 95℃ for 10 min, denaturation at 95℃ for 15 s, and extension at 60℃ for 1 min. MCMV qPCR data is represented as relative levels between groups, normalised to β-actin. Mice used were 7–10 weeks old.

## Flow cytometry

Cells were incubated for 10 min at 4℃ with Fc blocking antibody CD16/CD32 (2.4G2) and subsequently stained for 20 min at 4℃ with saturating concentrations of fluorophore conjugated antibodies. Antibodies used were as follows: NK1.1–eFluor 450 (PK136), NK1.1–BV421 (PK136), NK1.1-APC (PK136), NKp46–PerCP eFluor 710 (29A1.4), NKp46–PE (29A1.4), NKp46–efluor450 (29A1.4), CD3–FITC (145–2 C11), CD3–PacBlue (500A2), TCRβ–APC (H57-597), TCRβ–PE (H57-597), CD69–PerCp-Cy5.5 (H1.2F3), IFNγ–APC (XMG1.2), IFNγ–eFluor450 (XMG1.2), granzyme B–PE-Cy7 (NGZB), BrDU – APC (B44), TCRb–PECy7 (H57-597), CD69–PE (H1.2F3), NK1.1–APC (PK136), Ly49H – PE (3D10), CD11b-PE-Cy7 (M1/70), CD27- FITC (LG.7F9), PKM2- PE purchased from Abcam, eBiosciences, Biolegend, Thermo Fisher and BD Biosciences. L/D Aqua (Thermo Fisher) was used as a viability dye. Live cells were gated according to their forward scatter (FSC-A) and side scatter or according to L/D Aqua negative cells, single cells according to their FSC-W and FSC-A, NK cells were identified as NK1.1$^+$, NKp46$^+$ and CD3$^-$ cells. Cellular ROS measurements were obtained using H2-DCFDA flow

cytometric dye (5 µM) (Thermo Fisher). Cell proliferation experiments were carried out using Cell Trace Violet Cell Proliferation kit (Thermo Fisher). For intracellular staining, the cells were incubated for 4 hr with the protein transport inhibitor GolgiPlug (BD Biosciences). For fixation and permeabilisation of the cells, the Cytofix/Cytoperm kit from BD Biosciences was used according to manufacturer's instructions. BrDU staining was carried out according to manufacturer's instructions (BD Biosciences). Data were acquired on either a FACSCanto, a LSR Fortessa, or LSR II (Beckton Dickson). Flow cytometry data was analysed using FlowJo 10.

## Cytometric bead array

For cell supernatant measurements, cells were seeded at $2 \times 10^6$ per mL and treated with cytokines. After 18 hr, supernatants were harvested and frozen for later analysis. For serum measurements, blood was harvested from mice by cardiac puncture. Serum was harvested by centrifugation in serum separator tubes at top speed for two minutes. CBA was performed as per manufacturers (Becton Dickinson) instructions using 50 µL of supernatant or serum per sample and analysed by flow cytometry on a BD Fortessa. Data was analysed using FCAP Array software (BD Biosciences).

## Real-time quantitative PCR

Cultured NK cells were purified using MACS purification with the NK isolation kit II (Miltenyi Biotech) prior to stimulation. RNA was isolated using the RNeasy RNA purification mini kit (QIAGEN) or GeneJET RNA purification kit (Thermo Fisher)(according to the manufacturer's protocol). From purified RNA, complementary DNA (cDNA) was synthesised using the reverse-transcriptase kit qScript cDNA synthesis kit (Quanta Biosciences) or high-capacity reverse DNA synthesis kit (Applied biosciences). Real-time PCR was performed in triplicate in a 96-well plate using iQ SYBR Green-based detection on an ABI 7900HT fast qPCR machine. For the analysis of mRNA levels using SYBR green detection the derived values were normalised to Rplp0 mRNA levels. Primers: *Rplp0* forward: 5'-CATGTCGC TCCGAGGGAAG-3',*Rplp0* reverse: 5'-CAGCAGCTGGCACCTTATTG-3', *Pkm2* forward: 5'- GCTA TTCGAGGAACTCCGCC-3', *Pkm2* reverse:5'-AAGGTACAGGCACTACACGC-3'. For the analysis of mRNA levels using Taqman detection, the derived values were normalised to HPRT mRNA levels. *Pkm2:*Probe: 5'-/56FAM/TTATCGTTC/ZEN/TCACCAAGTCTGGCA/3IABkFQ /- 3',Primer 1: 5'TTCGAGTCACGGCAATGATAG-3', Primer 2: 5'-TCCTTCAAGTGCTGCAGTG-3', *Hprt:* Proprietary Probe Mm01545399_m1 Cat: 4331182.

## Proteomics

For proteomic analysis, $5 \times 10^6$ purified cultured NK cells were stimulated for 18 hr in RPMI media containing IL-2 (20 ng/ml) plus IL-12 (10 ng/ml). To remove dead cells, a density gradient (Lymphoprep, Axis-Shield) was used. Cells were spun down and stored at $-80°C$ until further preparation. Cell pellets were lysed in 400 µl lysis buffer (4% SDS, 50 mM TEAB pH 8.5, 10 mM TCEP). Lysates were boiled and sonicated with a BioRuptor (30 cycles: 30 s on, 30 s off) before alkylation with iodoacetamide for 1 hr at room temperature in the dark. The lysates were subjected to the SP3 procedure for protein clean-up before elution into digest buffer (0.1% SDS, 50 mM TEAB pH 8.5, 1 mM CaCl$_2$) and digested with LysC and Trypsin, each in a 1:50 (enzyme:protein) ratio. Tandem mass tag (TMT) labelling and peptide clean-up were performed according to the SP3 protocol. Samples were eluted into 2% dimethyl sulphoxide in water, combined and dried in vacuo. The TMT samples were fractionated using off-line high pH reverse phase chromatography: samples were loaded onto a $4.6 \times 250$ mm Xbridge BEH130 C18 column with 3.5 µm particles (Waters). Using a Dionex BioRS system, the samples were separated using a 25-min multistep gradient of solvents A (10 mM formate at pH 9 in 2% acetonitrile) and B (10 mM ammonium formate pH 9 in 80% acetonitrile) at a flow rate of 1 ml/min. Peptides were separated into 48 fractions which were consolidated into 24 fractions. The fractions were subsequently dried and the peptides redissolved in 5% formic acid and analysed by liquid chromatography–mass spectrometry (LC-MS). These data are publicly available at on the Immunological Protein Resource at immpres.co.uk.

## Western blotting

For western blot analysis, cells were harvested, washed twice with ice-cold PBS and lysed at $1 \times 10^7$/ ml in lysis buffer containing 50 mM Tris Cl pH 6.7, 2% SDS, 10% glycerol, 0.05% Bromophenol Blue,

1 µM dithiothreitol (DTT), phosphatase and protease inhibitors. Samples were denatured at 95°C for 10 min, separated by sodium dodecyl sulphate–polyacrylamide gel electrophoresis and transferred to a polyvinylidene difluoride membrane. PKM1 (D30G6), PKM2 (D78A4) and phospho- S6 ribosomal protein (Ser235/236) (D57.2.2E) were obtained from Cell Signalling. Actin (AC-15) was obtained from Abcam. Total S6 ribosomal protein (C-8) was obtained SantaCruz Biotechnology. Where western blotting stripping was required, Restore western blot stripping buffer (Thermo Fisher) was used, blots were re-blocked and subsequently probed for different proteins. For DSS crosslinking cells were suspended at $1 \times 10^7$/ml in PBS and treated with 500 µM DSS for 30 min at room temperature. After 30 min, 5 µl of 1M Tris pH 7.5 was added to quench unreacted DSS. Cells were then lysed in sample buffer and electrophoresed as described above.

## Seahorse metabolic flux analysis

For real-time analysis of the extracellular acidification rate (ECAR) and oxygen consumption rate (OCR) of purified and expanded NK cells cultured under various conditions, a Seahorse XF-24 Analyser or a Seahorse XFe-96e Analyser (Agilent Technologies) was used. In brief, 750,000 MACS purified, expanded NK cells were added to a 24-well XF Cell Culture Microplate, 200,000 MACS purified NK cells to a 96-well XFe Cell Culture Microplate. All cell culture plates were treated with Cell-Tak (BD Pharmingen) to ensure that the NK cells adhere to the plate. Sequential measurements of ECAR and OCR following addition of the inhibitors (Sigma) oligomycin (2 µM), FCCP (1 µM), rotenone (100 nM) plus antimycin A (4 µM), and 2-deoxyglucose (2DG, 30 mM) allowed for the calculation of basal glycolysis, glycolytic capacity, basal mitochondrial respiration, and maximal mitochondrial respiration.

## Pyruvate kinase activity assay

For analysis of pyruvate kinase activity, cultured NK cells were purified and stimulated and lysed using assay buffer from the manufacturer (Biovision). Calls were lysed as 75,000 cells per well and carried out in triplicate technical replicates. The assay was measured using absorbance over time on a SpectraMax plate reader.

## NADPH level assay

Cells were lysed in 0.1 M NaOH with 0.5% DTAC for NADPH determination at $0.1 \times 10^6$ cells per well. Relative NADPH levels were then determined using the NADPH-Glo assay kit (Promega) as per manufacturer's instructions.

## RNA sequencing

Two million MACS purified NK cells were washed twice in cold PBS. Cells were then lysed using the Gene Jet lysis buffer (Thermo Fisher). mRNA was purified using the Gene Jet RNA kit according to manufacturer's instructions. RNA was quantified using NanoDrop. RNA was then snap frozen in liquid nitrogen and sent on dry ice to the Frederick National Laboratory Sequencing facility for sequencing. RNA samples were first subjected to quality control analysis. RNA-Seq samples were then pooled and sequenced on HiSeq4000 using Illumina TruSeq Stranded Total RNA Library Prep and paired-end sequencing. The samples had 52 to 246 million pass filter reads with more than 90% of bases above the quality score of Q30. Reads of the samples were trimmed for adapters and low-quality bases using Cutadapt before alignment with the reference genome (Mouse - mm10) and the annotated transcripts using STAR. Library complexity was measured in terms of unique fragments in the mapped reads using Picard's Mark Duplicate utility. The gene expression quantification analysis was performed for all samples using STAR/RSEM tools.

For mRNA quantification, BAM files were imported into Partek Genomic Suite software (Partek Inc) and the built-in RNA-seq workflow pipeline was used. Reads were aligned and quantified using RefSeq transcripts database based on the E/M algorithm.

Gene-level RPKMs were used for subsequent analyses. Differentially expressed genes were identified by analysis of variance (ANOVA). The threshold value has been set at 1.5-fold change and a false discovery rate (FDR) < 0.05. Differentially expressed genes were analysed by ingenuity pathway analysis (IPA; Ingenuity Systems).

## Metabolomics

### GC-MS metabolomics

Untargeted metabolomics was carried out by West Coast Metabolomics at UC Davis. $10 \times 10^6$ MACS purified NK cells were washed three times in cold PBS. Samples were then snap frozen in liquid nitrogen. At UC Davis, samples were re-suspended with 1 ml of extraction buffer (37.5% degassed acetonitrile, 37.5% isopropanol and 20% water) at –20°C, centrifuged and evaporated to complete dryness. Membrane lipids and triglycerides were removed with 50% acetonitrile in water. The extract was aliquoted into two equal portions and the supernatant evaporated again. Internal standards C08-C30 fatty acid methyl esters were added and the sample derivatised by methoxyamine hydrochloride in pyridine and subsequently by N-methyl-N-trimethylsilyltrifluoroacetamide for trimethylsilylation of acidic protons. Gas chromatography-time-of-flight analysis was performed by the LECO Pegasus IV mass spectrometer. Samples were additionally normalised using the sum of peak heights for all identified metabolites (mTIC Normalisation).

### LC-MS metabolomics

Cell pellets for targeted analysis were washed and resuspended in ice cold 80% methanol. Phase separation was achieved by centrifugation at 4°C and the methanol-water phase containing polar metabolites was separated and dried using a vacuum concentrator. The dried metabolite samples were stored at −80°C and resuspended in Milli-Q water the day of analysis. An Agilent 6410 Triple Quadrupole mass spectrometer interfaced with a 1200 Series HPLC quaternary pump (Agilent) was used for ESI-LC–MS/MS analysis in multiple reaction monitoring mode. Seven concentrations of standards, processed under the same conditions as the samples, were used to establish calibration curves. The best fit was determined using regression analysis of the peak analyte area. Chromatographic resolution was obtained in reverse phase on a Zorbax SB-C18 (1.8 μm; Agilent) for amino acids and an Eclipse Plus C18 (1.8 μm; Agilent) for TCA and PPP intermediates, with a flow rate set at 0.4 ml/min. Data were normalised to protein concentration.

## Statistical analysis

Statistical analysis was performed by GraphPad Prism 8, with the tests used indicated in the figure legend. Datasets where two independent parameters were being compared (genotype and stimulation/treatment) were analysed by two-way ANOVA with Sidaks post-test. Datasets one variable parameter (treatment) with more than two groups were analysed by one-way ANOVA with Tukey's post-test. Datasets with one variable (treatment) and two groups were analysed by a Students t test. Data were log transformed where appropriate before analysis. Flow cytometry data was analysed using FlowJo 10. $*p<0.05$, $**p<0.01$, $***p<0.001$. All error bars represent the mean ±the standard error of the mean (S.E.M).

## Acknowledgements

JFW was funded, in part, by a Wellcome Trust PhD Studentship (106811/Z/15/Z). This research was supported in part by the Intramural Research Program of the NIH, National Cancer Institute, Center for Cancer Research and funding from Scientific Foundation Ireland (18/ERCS/6005).

## Additional information

### Funding

| Funder | Grant reference number | Author |
| --- | --- | --- |
| National Institutes of Health | | Jessica F Walls<br>Jeff J Subleski<br>Erika M Palmieri<br>Marieli Gonzalez Cotto<br>Daniel W McVicar |
| Wellcome | 106811/Z/15/Z | Jessica F Walls |
| Science Foundation Ireland | 18/ERCS/6005 | David K Finlay |

The funders had no role in study design, data collection and interpretation, or the decision to submit the work for publication.

## Author contributions
Jessica F Walls, Formal analysis, Investigation, Methodology, Writing - original draft, Writing - review and editing; Jeff J Subleski, Methodology, Writing - review and editing; Erika M Palmieri, Investigation, Writing - review and editing; Marieli Gonzalez-Cotto, Formal analysis, Writing - review and editing; Clair M Gardiner, David K Finlay, Conceptualization, Supervision, Methodology, Writing - original draft, Writing - review and editing; Daniel W McVicar, Conceptualization, Resources, Supervision, Funding acquisition, Methodology, Writing - original draft, Writing - review and editing

## Author ORCIDs
Jessica F Walls ⬤ https://orcid.org/0000-0003-2901-3318
Erika M Palmieri ⬤ https://orcid.org/0000-0002-4237-5324
Marieli Gonzalez-Cotto ⬤ https://orcid.org/0000-0003-3359-8814
David K Finlay ⬤ https://orcid.org/0000-0003-2716-6679

## Ethics
Animal experimentation: Mice utilised in Ireland were maintained in compliance with Irish Department of Health and Children regulations and with the approval of the University of Dublin's ethical review board. Mice utilised in the USA were maintained in accordance with institutional guidelines for animal care and use at NCI Frederick, NIH.

## Decision letter and Author response
Decision letter https://doi.org/10.7554/eLife.59166.sa1
Author response https://doi.org/10.7554/eLife.59166.sa2

# Additional files
## Supplementary files
• Supplementary file 1. $Pkm2^{NK-WT}$ and $Pkm2^{NK-KO}$ cultured and purified cells were stimulated with (IL-2/12) or left unstimulated (low-dose IL-15) for 18 hr. HiSeq RNA sequencing was then performed. Differential gene expression analysis was carried out using Partek to assess total gene changes between IL-2/12 stimulated $Pkm2^{NK-WT}$ and $Pkm2^{NK-KO}$ cells. Total gene changes were assessed at a fold change cut-off of 1.5 and a p value of 0.05 with an FDR of 0.05.

• Supplementary file 2. Cultured wildtype NK cells were stimulated with IL-2/12 +/- TEPP-46/vehicle for 18 hr. HiSeq RNA sequencing was then performed. Differential gene expression analysis was carried out using Partek to assess total gene changes between IL-2/12 TEPP-46 (50 µM) and IL-2/12 Vehicle (0.1% v/v DMSO). Total gene changes were assessed at a fold change cut-off of 1.5 and a p value of 0.05 with FDR 0.05.

• Transparent reporting form

## Data availability
RNA sequencing data has been uploaded to GEO (GSE156064).

The following dataset was generated:

| Author(s) | Year | Dataset title | Dataset URL | Database and Identifier |
|---|---|---|---|---|
| Walls JF, Subleski JJ, Palmieri EM, Gonzalez-Cotto M, Gardiner CM, McVicar DW, Finlay DK | 2020 | Total RNA sequencing of cultured splenic PKM2WT and PKM2KO NK cells | http://ncbi.nlm.nih.gov/geo/query/acc.cgi?acc=GSE156064 | NCBI Gene Expression Omnibus, GSE156064 |

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
