## [Decision Letter]

**Acceptance summary:**

This study investigates the role of a glycolytic enzyme called PKM2 in regulating the functions of natural killer cells. The authors show that PKM2 tetramerization during NK cell activation regulates PKM2 activity to influence NK cell growth, metabolism, and function.

**Decision letter after peer review:**

Thank you for submitting your article "Metabolic but not transcriptional regulation by PKM2 is important for Natural Killer cell responses" for consideration by *eLife*. Your article has been reviewed by three peer reviewers, one of whom is a member of our Board of Reviewing Editors, and the evaluation has been overseen by Tadatsugu Taniguchi as the Senior Editor. The following individual involved in review of your submission has agreed to reveal their identity: Ruoning Wang (Reviewer #3).

The reviewers have discussed the reviews with one another and the Reviewing Editor has drafted this decision to help you prepare a revised submission.

Summary:

In this study, Walls et al. examine the role of PKM2 in NK cells. PKM2, the glycolytic enzyme that converts PEP to pyruvate, is expressed in NK cells and further upregulated during NK cell activation by Il-2/IL-12 stimulation. However, NK cells lacking PKM2 are activated normally in vitro and in vivo during MCMV infection, as indicated by proliferation, production of IFNγ and TNF, expression of Granzyme B, and viral clearance. The authors attribute the lack of phenotype to compensatory induction of PKM1. The authors' findings also suggest that while in other cell types PKM2 may "moonlight" in a transcriptional role, any such role for PKM2 in NK cells seems not to influence NK cell activation, at least in the contexts studied.

PKM2, unlike PKM1, can form a tetramer with increased enzyme activity. In other contexts, such tetramerization is thought to enhance flux through glycolysis which disfavors glycolytic intermediates from being diverted to biosynthetic shunts like the PPP. The authors next asked how such tetramerization of PKM2 may influence NK cell activation, using a small molecule TEPP-46 that enhances PKM2 tetramerization. The authors found that TEPP-46 treatment during NK cell activation led to reduced cellular growth, reduced production of the PPP metabolites R5P and NADPH, increased cellular ROS, and reduced oxidative metabolism, as well as reduced production of IFNγ and TNF and reduced expression of Granzyme B.

Essential revisions:

1) The authors should provide some mechanistic insight into how PKM2 tetramerization leads to reduced NK cell activation. Does treatment with ROS scavengers like NAC or cell permeable glutathione rescue the effects of TEPP-46 on NK cell activation?

2) Does PKM2 undergo tetramerization in a physiological context? Given the lack of a phenotype in the PKM2 KO in the in vitro or in vivo conditions that the authors analyzed, it seems like tetramerization may not occur (because PKM1, which is upregulated, is thought to not tetramerize). At the very least, the authors should discuss under what conditions PKM2 tetramerization can occur to suppress NK cell activation.

3) The authors should confirm that TEPP-46 has no effect in PKM2 KO cells.

---

## [Author Response]

Essential revisions:1) The authors should provide some mechanistic insight into how PKM2 tetramerization leads to reduced NK cell activation. Does treatment with ROS scavengers like NAC or cell permeable glutathione rescue the effects of TEPP-46 on NK cell activation?

We have looked into possible mechanisms involved and have found that ROS scavengers can rescue IFNγ production in TEPP-46 treated NK cells. TNF and IL-10 could not be rescued by NAC treatment.

We also find that TEPP46 treatment reduces the protein expression of cMyc in NK cells, a transcription factor that has shown to be required for NK cell functional responses by our previous study (Loftus et al., 2018) (Figure 8K). However this altered cMyc expression is not strongly reflected in the transcriptomic dataset when using a fold change cut off of 2. Interestingly, we show that the ROS scavenger Mitoquinol rescued the protein expression of cMyc in TEPP-46 treated NK cells. These data have now been included in in Figure 9 and the Discussion has been modified.

2) Does PKM2 undergo tetramerization in a physiological context? Given the lack of a phenotype in the PKM2 KO in the in vitro or in vivo conditions that the authors analyzed, it seems like tetramerization may not occur (because PKM1, which is upregulated, is thought to not tetramerize). At the very least, the authors should discuss under what conditions PKM2 tetramerization can occur to suppress NK cell activation.

This is indeed a valid question. The relative biological benefit of having either mono/dimeric or tetrameric PKM2 has been of much interest to us. We have now included a crosslinked western blot showing that IL-2/12 stimulated NK cells have both monomeric and tetrameric PKM2 (Figure 1E) levels of which could be adjusted when the cells are in different anabolic or catabolic states. As demonstrated in this manuscript, too much pyruvate kinase activity is detrimental to the cell and leads to oxidative stress making regulation essential.

3) The authors should confirm that TEPP-46 has no effect in PKM2 KO cells.

Thanks, this is another important comment. We have done this and can confirm that TEPP-46 does not affect PKM2-KO NK cells with respect to IFNγ production and granzyme B expression. We also ensured that TEPP-46 does not increase pyruvate kinase activity in PKM2-KO NK cells, confirming that it has specificity for PKM2 and not PKM1. These data are now included as Figure 6—figure supplement 1.